
# Evaluation of error components in rainfall retrieval from collocated commercial microwave links

Anna Špačková[1], Martin Fencl[1], Vojtěch Bareš[1]

[1]Dept. of Hydraulics and Hydrology, Czech Technical University in Prague, Prague, 166 29, the Czech Republic

*Correspondence to*: Anna Špačková (anna.spackova@cvut.cz)

**Abstract.** Opportunistic rainfall sensing using commercial microwave links (CMLs) operating in telecommunication networks has the potential to complement conventional rainfall monitoring, however, the diversity of sensors and their errors are difficult to handle. This analysis empirically evaluates errors of CML observations that manifest discrepancies between collocated

sensors without reference rainfall measurements. Collocated CMLs are evaluated as independent rainfall sensors, which enable us to assess the effect of hardware homogeneity and measurement consistency using CML observations at twelve sites within a real telecommunication network in Prague. The evaluation considers 33 rainfall events distinguishing between stratiform and convective rainfall type in the period 2014 and 2016, monitored in 1-min temporal resolution. Collocated CMLs of identical and different frequencies are evaluated, and different rainfall types are discussed. The collocated commercial microwave links

are in excellent agreement. The inherent error of rain-induced attenuation for paired independent commercial microwave links is 0.4 dB. The high correlation of the rainfall intensity measurements between the collocated sensors was obtained in a range between 0.96 and 0.99, and the root mean square error ranges between 0.4 mm h$^{-1}$ and 1.7 mm h$^{-1}$. This confirms homogeneous behaviour of the hardware in a real network. Therefore, the data of CMLs of the same characteristics can be processed with identical parameters for rainfall retrieval models.

## 1 Introduction

Commercial microwave links (CMLs) are appealing opportunistic rainfall sensors which have the great potential to complement traditional rain gauge networks (Graf et al., 2020) and to provide rainfall observations in sparsely gauged regions (Walrefen et al., 2022). CMLs are point-to-point radio connections used worldwide as the backhaul of telecommunication networks. When it rains, the CML radio connection is attenuated by raindrops. Rainfall estimation takes advantage of the

nearly proportional relation between raindrop attenuation and rain rate along a CML path at frequencies typically used by cellular operators.

Several studies demonstrated the potential of CMLs for different hydrometeorological applications. For example, Bianchi et al. (2013) assimilated them with radar observations, Graf et al. (2020) reconstructed German-wide rainfall maps,



and Imhoff et al. (2020) used CMLs for nowcasting. Locally, Pastorek et al. (2021) demonstrated that CMLs can conveniently

complement relatively dense rain-gauge networks and can be successfully used for rainfall-runoff modelling.

CML networks are substantially denser in comparison to rain-gauge networks operated by national meteorological offices in developed countries (van de Beek et al., 2020) and the coverage extends well into emerging countries which have sparse conventional monitoring networks (Doumounia et al., 2014). CMLs provide path-averaged measurements which can be acquired online from a network operation centre with a delay of only a few seconds (Chwala et al., 2016). Compared to

weather radar, they usually measure rainfall only several tens of meters above ground and the relation between attenuation, and rain rate is less sensitive to drop size distribution than the Z-R relationship (Berne and Uijlenhoet, 2007).

On the other hand, there are also many challenges related to CML rainfall estimation. One of the main challenges of opportunistic sensors is the lack of control over them, including incomplete metadata about their hardware and operation (e.g., de Vos et al., 2019). CMLs are not an exception as telecommunication networks are formed by a variety of devices which

affect the homogeneity of attenuation observations. This may lead to different requirements on raw data processing for individual CMLs and differences in error types, as well as their magnitudes. In addition, even the same type of hardware does not ensure consistent performance in terms of rainfall estimation as CMLs are primarily telecommunication devices and their design, deployment, and maintenance are exclusively focused on their primary function.

The models used to separate raindrop path attenuation from other attenuation components rely on empirically esti-

mated parameters. If, for example, wet antenna attenuation (discussed in greater detail in Section 2) differs significantly among CMLs, extensive calibration campaigns are required to obtain accurate rainfall estimates from the network of CMLs. High requirements on CML calibration disables both the upscaling of rainfall estimation, as well as the application in sparsely gauged regions where rainfall information from opportunistic sensors, such as CMLs, have the greatest added value. Understanding the effects of hardware heterogeneity on CML rainfall retrieval is also crucial when estimating the uncertainties of

retrieved CML rain rates. Spatially inconsistent errors make it difficult to interpolate or reconstruct gridded rainfall fields from CML observations, and hinder assimilation with other rainfall observations (Graf et al., 2021).

Until now, studies evaluating CML reliability have focused on comparisons with traditional observations used as reference, although the path-integrated character of CML observations makes such comparisons difficult. Experimental setups with a single CML obtain reference rainfall observation from several rain gauges or disdrometers along a CML path (Leth et

al., 2018; Špačková et al., 2021; Moroder et al., 2020). Gathering such high-quality reference data is not feasible at the scale of the entire network or its parts. In these cases, gauge-adjusted radar is the most used reference (Graf et al., 2020; Overeem et al., 2013). Such comparisons are affected by substantial errors, especially during heavier rainfalls (Schleiss et al., 2020) and at sub-hourly time scales (Ochoa-Rodriguez et al., 2019). Studies primarily evaluating consistency between rain rate estimates of different CMLs have not been, to the best of our knowledge, reported.

Inconsistency between rain rate observations also occurs by traditional monitoring networks as they evolve over time and rarely consist of the same devices. Studies evaluating observations from collocated devices are often used to investigate



discrepancies between different types of instruments. For example, their results are used for homogeneity adjustments of atmospheric climate data (e.g., Peterson et al., 1998). Collocated devices of the same type, where none of the measurements are considered to be error prone references, are also used to calibrate error models. For example, Ciach and Krajewski (2005) or

Peleg et al. (2013) used pairs of rain gauges located at different sites when investigating the small-scale variability of rainfall to quantify the nugget of variograms.

In this paper, we investigate the consistency between different CMLs by comparing pairs of independent collocated devices operating at the same frequencies and at different frequencies. We quantify the discrepancies between their observations and identify the random and systematic components of these discrepancies. Finally, we assess the effect of CML pro-

cessing on the measurement consistency of collocated pairs. Twelve sites of collocated CMLs in the Prague CML network, having almost identical paths and thus being exposed to the same atmospheric conditions, were selected for the experiment. The analysis is based on observations during 33 rainfall events over a three-year period.

The paper is structured as follows: Section 2 elaborates on CML rainfall retrieval and sources of associated uncertainties in the rainfall estimates. The dataset and methodology used in this study are described in Section 3. Section 4 presents

the results which are further discussed in Section 5, and, finally, the conclusions are presented in Section 6.

## 2 CML rainfall retrieval and its uncertainty

CML rainfall retrieval utilizes significant attenuation of electromagnetic (EM) waves by raindrops at millimetre wavelengths (ITU-R, 2005). Raindrop path attenuation is not directly observed but estimated from the total loss, typically as the difference between its level during rainy and dry weather, and accounting for wet antenna attenuation (WAA) that is caused by water

film forming on antenna radomes during rain and dew events (Chawla and Kunstmann, 2019). The specific raindrop attenuation (dB km$^{-1}$) is then converted to the rain rate. The reliability of CML rainfall estimation depends on the accuracy and precision of measured transmitted ($TSL$) and received signal powers ($RSL$), the efficiency of separating raindrop attenuation from other losses, the accuracy of the attenuation-rain-rate relation, and the overall sensitivity of a CML to raindrop path attenuation.

Nominal values of $TSL$ and $RSL$ accuracies are usually unknown, nevertheless, maximal precision is given by the

quantization of $TSL$ and $RSL$ records. Based on the hardware, the power levels are often provided at 0.1, 0.3 or 1.0 dB (Chwala and Kunstmann, 2019). Most modern CMLs feature automatic transmit power control (ATPC), which adjusts the power level of transmitted EM waves so that $RSL$ is maintained at a specific level, or in some predefined range. In the first case, $TSL$ changes even with a small increase or decrease of $RSL$, whereas in the second case, $TSL$ remains constant in some wider range of $RSL$ values (e.g., 5 dB). When the quantization of $RSL$ and $TSL$ differs, the configuration of the ATPC affects the

precision of the calculated total loss ($TSL - RSL$). For example. Ericsson MINILINK CMLs provide $RSL$ and $TSL$ readings with a quantization of 0.3 resp. 1 dB. In the first case, where $TSL$ changes instantly, the quantization of total loss is practically only 1 dB, whereas in the second case, the quantization is 0.3 dB.



Total loss $L_t$ (dB), which is the difference between the transmit and received signal powers, consists of several components as described below:

$$L_t = TSL - RSL = L_{bf} + L_m + L_{tc} + L_{rc} - G_t - G_r, \tag{1}$$

where $L_{bf}$ is free space loss, $L_m$ are losses in the medium, $L_{tc}$ and $L_{rc}$ are losses in the transmitting and receiving antennas and $G_t$ and $G_r$ are gains of transmitting resp. receiving antennas. The free space loss is uniquely defined by the distance between the transmitter and receiver and by the wavelength. Losses in the medium consist of gaseous attenuation, losses due to obstacles in the path, diffraction losses causing bending of a direct wave towards ground, and losses due to raindrops along the CML path. The sum of losses and gains is determined by the CML hardware and includes wet antenna attenuation (WAA). WAA is an attenuation caused by the formation of wetness on an antenna radome which is a complex process influenced by atmospheric conditions and hydrophobic properties of the radome surface (Leth et al., 2018).

Rain-induced attenuation $Ar$ (dB) is expressed as total loss $L_t$ (dB) with separated baseline attenuation $B$ (dB). Baseline attenuation is usually estimated from attenuation during dry-weather conditions:

$$Ar = L_t - B \tag{2}$$

An important part of most baseline estimation methods is, thus, the classification of dry and wet weather from CML data (Chwala and Kunstman, 2019). Baseline methods implicitly assume that components of total loss, other than raindrop attenuation and WAA, are identical during dry and wet weather. This is not necessarily fulfilled as water vapor content in the air, as well as temperature, can differ between dry and wet weather periods (Minda and Nakamura, 2005). This inevitably leads to uncertainties in the estimated baseline and, thus, in estimated raindrop path attenuation. Furthermore, multipath inferences, diffraction losses (Valtr et al., 2011), or attenuation due to dust (Abuhdima and Saleh, 2010) and fog (David et al., 2009) can affect the accuracy of the estimated baseline.

The specific raindrop path attenuation $k$ (dB km$^{-1}$) is usually obtained by a data-driven approach, separating wet antenna attenuation $Aw$ (dB) from rain-induced attenuation $Ar$ (dB):

$$k = \max\left(\frac{Ar - Aw}{l}; 0\right), \tag{3}$$

where $l$ is CML path length (km). In practice, WAA is either assumed to be constant (Overeem et al., 2011), or it is estimated as a time-dependent process (Schleiss et al., 2013), or a function of rain rate or attenuation (Leijnse et al., 2008; Kharadly and Ross, 2001). For an intercomparison of different models see (Pastorek et al., 2022). The variety of approaches reported in the literature indicates that WAA magnitude and pattern differ among CMLs. Therefore, WAA represents one of the most severe sources of uncertainty.

The relation between specific raindrop attenuation $k$ (dB km$^{-1}$) and rain rate $R$ (mm h$^{-1}$) is approximated by a power law:

$$R = \alpha\, k^{\beta}, \tag{4}$$





where $\alpha$ (mm h$^{-1}$ dB$^{-\beta}$ km$^{\beta}$) and $\beta$ (-) are empirical parameters dependent on CML frequency, polarization, and drop size
distribution (DSD). When estimating rainfall, alpha and beta parameters are either taken from ITU recommendations (ITU-R, 2005) or estimated from local disdrometer observations (e.g., Schleiss et al., 2013). In both cases, one set of parameters is used for all rainfall types. Furthermore, uniform rain rate is assumed along a CML path. The uncertainties related to variable DSD in real rainfalls are lowest for frequencies between 15 and 40 GHz (with a minimum at 30 GHz), where $\beta$ is close to unity (Berne and Uijlenhoet, 2007) but increases outside of this range.

The magnitude of total rainfall estimation error and the ratio between its components is strongly related to CML sensitivity to raindrop path attenuation, which is given by CML path length, frequency, and polarization. For example, the limited precision of $TSL$ and $RSL$ readings or incorrectly estimated WAA results in larger errors by CMLs relatively insensitive to rainfall. On the other hand, errors related to $k$-$R$ relation non-linearity can be more pronounced by longer CMLs, whereby rain rates along their path are more likely to be non-uniform. Diversity in CML length and frequencies common to standard cellular networks can help to estimate the contribution of different errors to the total measurement error and possibly reduce some systematic deviations. For example, Fencl et al. (2019) used extremely short CMLs insensitive to raindrop path attenuation to quantify WAA. The same collective of authors quantified WAA during light rainfalls at E-band CMLs by comparing observations of CMLs having different path lengths (Fencl et al., 2020). Collocated CMLs, investigated in this paper, could further improve error quantification, especially in cases where CML observations are available alone.

Collocated CMLs of the same frequency and polarization must be affected by the same free space losses ($L_{bf}$) and losses in the medium ($L_m$) as their paths are identical and their EM waves are propagated through the same medium. The gains ($G_t + G_r$) and the losses ($L_{tc} + L_{rc}$) on the antennas of collocated CMLs can, however, differ due to the hardware used. Losses on the antennas also include WAA caused by the formation of a water layer on the antenna radomes. Droplet patterns can substantially differ between the antennas depending on the material of a radome and on the condition of its hydrophobic coating (Leth et al., 2018). On the other hand, antenna radomes are exposed to similar rain rates and other atmospheric conditions by collocated CMLs.

EM waves of collocated CMLs operating at different frequencies, and/or polarizations, are propagated by the same medium, however, they interact differently with gasses or droplets of different sizes. Discrepancies in their rain rate estimates thus combine hardware-related errors with errors related to the propagation through the medium.

This paper examines collocated CMLs having the same frequency and different frequencies. They are affected by the same weather conditions along their path. Discrepancies between collocated CMLs of the same frequency are only due to hardware characteristics and enable us to evaluate the effect of hardware homogeneity. Considering CML pairs of different frequencies, the discrepancies are due to hardware inhomogeneity combined with a different interaction of EM waves to the propagation medium.



## 3 Materials and methods

This study investigates observations of collocated independent CMLs at twelve sites located in Prague, Czech Republic (Fig. 1) for 33 rainfall periods (304 h) in the non-winter periods of the years 2014 to 2016. There are eight sites of CML pairs operating at identical frequency bands (indicated by "a" in the site name) and six sites of CML pairs operating at different frequency bands (indicated by "b" in the site name), which implies that sites "b" contain collocated CML pairs of similar lengths, but of different frequencies. Sites 2 and 3 offer pairs of both, one pair of identical and one pair of different frequency bands (Fig. 1).

First, rain-induced attenuation is intercompared for sites with CML pairs operating at identical frequency bands. Second, the retrieved collocated rainfall intensity is evaluated for all sites. For rainfall retrieval, the WAA model (Pastorek et al., 2022) optimized to rain gauge adjusted weather radar rainfall observations is used.

### 3.1 Materials

#### *CML*

The collocated CMLs share one end-point (e.g., the same rooftop) and the distance between the second end-points is not greater than 10 % of the mean path lengths of the collocated CMLs. The maximum distance between the end nodes of the CMLs is 0.51 km, and occurs at site 12b which contains the longest CMLs. To compare, such a distance is approximately half of a standard weather radar pixel. Most of the link paths cover urbanized areas. The CMLs operate at frequency bands 22, 25, 32 and 38 GHz and have lengths of 682 m to 5836 m (Fig. 2).

The CMLs are part of telecommunication backhaul operated by T-Mobile Czech Republic a.s. which employs an Ericsson MINILINK platform (except for one Bosch Marconi PS UHP antenna at site 7a) featuring a duplex configuration with two sublinks having frequency separation close to 1 GHz. The antenna radomes are 0.3 m or 0.6 m in diameter. All the metadata, including polarizations, are summarized in Appendix A. The transmitted and received data are collected by custom-made data acquisition software. The software polls attenuation data approximately every 10 s from both sublinks of a single CML. *TSL* and *RSL* have a quantization of 1 dB and 1/3 dB respectively.

#### *Gauge-adjusted radar rainfall*

Adjusted weather radar observations are used for the optimization of the WAA model proposed by Pastorek et al. (2022) and for a visual classification of rainfall types.

Tipping bucket rain gauges are used for the adjustment of radar data. The Prague permanent municipal RG network consists of 23 stations, operated and maintained by Pražské vodovody a kanalizace, a.s. Tipping bucket rain gauges have a sampling area of 500 cm$^2$ and a resolution 0.1 mm (MR3, METEOSERVIS v.o.s.). The rain gauges are maintained monthly and regularly dynamically calibrated. The recorded tips are transformed to 1-min rainfall intensity data.

Raw radar images are composites of two dual-polarized weather radars (CZRAD) operated by the Czech Hydrometeorological Institute (CHMI). Radar reflectivity is taken from an altitude of 2 km above sea level (CAPPI 2 km). The composite product has a spatial and temporal resolution of 1 km × 1 km, resp. 5 min.





## 3.2 Rainfall events

This study uses a data set collected during the monitoring of the period between July 2014 and October 2016. Winter periods are excluded as the CML signal would be attenuated by solid precipitation. In total 33 rainfall events (see Appendix B and Figure 3) with a safety window of ± 5 h were selected. Event durations range from one hour to 1.5 days, which provides in total 304 h and 15 min (of which 153 h are convective events and 151 h and 15 min are stratiform events). Both stratiform events, with lower intensities having more uniform structures, and convective events, with spatially limited area and greater maximal intensities, are present in the collected data.

From a visual inspection of weather radar images, events were divided into two groups: First, convective rainfall events characterized by higher intensities, short durations, and a spatially limited area. Second, stratiform rainfall events with lower intensities, longer duration, and a more extensive area. There were ten stratiform events and 23 convective events, of which 21 were in the spring and summer seasons. During each event, the maximal 5-min rainfall intensity was calculated for each of the 23 municipal rain gauges, as well as the median of these intensities (Figure 3). The median maximal intensity during convective events reached up to 68 mm h$^{-1}$ and during stratiform events up to 21 mm h-1. The median durations of convective and stratiform events was 4 h (in the range of 1 to 18 h), resp. 12 h (in the range of 5 to 34 h).

## 3.3 Processing of raw CML data

First, total losses are calculated as the difference between $TSL$ and $RSL$ (Eq. 1) and aggregated to 1-min temporal resolution using averaging. The data undergo a visual inspection of hardware related errors and erratic behaviour. Issues, such as degraded sublink resolution, sudden change of baseline, and constant signal level during a rainfall event, are flagged (similar to Fencl et al., 2020) and removed from the processing.

Baseline $B$ for a certain timestep $n$ is estimated as a centred 10-day moving average of the total loss:

$$B_n = \frac{(L_t)_{n-p} + 2\,(L_t)_{n-p+1} + \dots + 2\,(L_t)_n + \dots + 2\,(L_t)_{n+p-1} + (L_t)_{n+p}}{4\,p}, \tag{5}$$

where $n$ is the timestep of interest (the centre of the 10-day period) and $p$ is the number of timesteps in a 5-day period (half of the 10-day period).

Further, a wet antenna attenuation model was used and explicitly derived from rainfall intensity $R_{est}$, which is not available directly, but is estimated from observed attenuation (Pastorek et al., 2022):

$$Aw = C\,(1 - exp(-d\,R_{est}^{\,z})), \tag{6}$$

where $C$ (dB) is the maximum $Aw$ and $d$ and $z$ are power law parameters. Parameter $d$ is 0.1. Parameters $C$ and $z$ are optimized by minimizing the difference between CML rain rates with WAA correction and reference rain rates along the CML path obtained from gauge-adjusted weather radar. Reference rain rates were calculated as a mean of rain rates along a CML



path weighted by a CML path length intersecting radar product grid cells. The parameters for all links are optimized to minimum root mean squared error (RMSE) between CML and radar rain rates at hourly timesteps. The optimization uses complete hourly observations.

The adjusted radar rainfall information is retrieved as follows. Radar reflectivity $Z$ (mm$^6$ m$^{-3}$) is transformed to rainfall intensity $R$ (mm h$^{-1}$) via the Marshall-Palmer relationship (Marshall and Palmer, 1948):

$$Z = a_{rad}\,R^{b_{rad}}, \tag{7}$$

with parameters $a_{rad}$ = 200 and $b_{rad}$ = 1.6. In addition, two threshold filters are applied (Novák and Sokol, 2008): (i) a filter for the reduction of weak non-precipitation echoes: $R$ ($Z$ < 7 dBZ) = 0 mm h$^{-1}$; and (ii) a filter for the reduction of rainfall overestimation caused by hail in convective storms: $R$ ($Z \geq$ 55 dBZ) = 99.85 mm h$^{-1}$.

A 35 km × 35 km area covering Prague was selected from raw radar rainfall images (750 km × 550 km). The data were resampled to 1 h timesteps. Multiplicative bias correction was performed using the wradlib Python package (Heistermann et al., 2013) and data from 23 municipal rain gauge stations.

Having estimated both baseline attenuation and WAA, the specific attenuation $k_{SUB}$ (dB km$^{-1}$) can be calculated using Eq. 2 and 3. Rainfall intensity is calculated for individual sublinks using Eq. 4 with parameters taken from ITU recommendations (ITU-R, 2005). The mean of the rainfall intensity of the two sublinks is used for the evaluation of the collocated CMLs:

$$R_{CML} = \frac{R_{SUB,1} + R_{SUB,2}}{2} \tag{8}$$

In addition to the rainfall retrieval procedure described above (Eq. 1-8, referred to as method 0) three other modifications of the procedure were tested and evaluated on a one-month-long subset of the data. They differ in the algorithm used for the baseline and/or WAA estimation:

- The baseline estimated as a monthly 1% quantile of the total losses and a constant WAA (1.5 dB) separation model (referred to as method 1).
- The constant baseline model based on the preceding dry period (a centred 60-min window of rolling standard deviation suggested by Schleiss and Berne (2010)) and the semiempirical WAA model proposed by Leijnse et al. (2008) for WAA (referred to as method 2).
- The baseline is the same as the previous in combination with the time-dependent WAA model with parameters suggested by Schleiss et al. (2013) (referred to as method 3).

**3.4 Quantitative evaluation**

CML measurement consistency is evaluated by comparing the observations of the CML pairs (Figure 2) during 33 events.

First, rain-induced attenuations of collocated CMLs operating at identical frequency bands are compared. The mean attenuation of CML sublinks is used for this purpose:



$$Ar_{CML} = \frac{Ar_{SUB,1} + Ar_{SUB,2}}{2}, \qquad (9)$$

Second, the estimated rain rates from the collocated CMLs are compared. Sites with identical and different frequency band pairs are evaluated separately. Third, the effect of time aggregations (1, 5, 15 and 30 min) on the consistency of CML rain rates is evaluated, and fourth, the evaluation is performed separately for events of different rainfall type.

Data visualisations via scatterplots are complemented with metrics for the overall evaluation: Pearson correlation coefficient $r$ (hereafter referred to as correlation), and root mean squared error (RMSE). Furthermore, double-mass curves and relative error are used. All timesteps in which one of the CMLs in the pair is lacking data are excluded from the evaluation.

The correlation expresses the level of the linearity relation between collocated CML observations. As none of the collocated observations can be considered as a true reference, RMSE and relative error reflect only the agreement of two measurements and are quantified with respect to one CML from the pair, which is further referred to as CML 1. In the case of the "b" sites, CML 1 is always the CML with the lower frequency in the pair. In this context, the RMSE reflects the deviation or random component rather than an error from the truth. Similarly, the relative error and double-mass curves assess systematic deviations between the CMLs. The double-mass curve compares rainfall cumulation of two datasets and is used to observe the consistency of the measurements during observation period.

## 4 Results

In this section, the results of the analysis on CML pairs operating at identical and different frequency bands, the effect of rainfall type and time aggregation, and the outcomes of different processing methods are presented.

### 4.1 Evaluation of CML pairs operating at identical frequency bands

Eight sites contain CMLs operating in pairwise-identical frequency bands are evaluated in this subsection (Fig. 1 and Fig. 2).

Figure 4 demonstrates a single event with excellent agreement between two collocated CMLs at two stages of processing: rain-induced attenuations are shown in the top panel, and rain rates in the bottom panel. There is clear similarity in the timeseries of the collocated sensors for both rain-induced attenuation and rainfall intensity. However, relatively higher differences can be seen for lower rainfall intensities, which, on the other hand, is not evident from the scatterplots (Fig. 6).

Figure 5 represents all eight CML pairs and shows a high correlation of 0.98 and low RMSE of 0.4 dB of rainfall induced attenuation, on average, for all pairs of CMLs. Within the eight pairs, the correlation ranges between 0.96 and 0.99 and RMSE between 0.2 and 0.5 dB. Overall, RMSE in rain-induced attenuation is 0.4 dB, which is close to signal quantization. The effect of the hardware on observed attenuation is similar within the CML pairs. To conclude, this indicates that the component of random error is low in total CML observation error.

Scatterplots and performance measures of rainfall intensities confirm the small variability of independent collocated measurements (Fig. 6). The correlation ranges from 0.97 to 0.99 and RMSE is between 0.4 and 0.9 mm h$^{-1}$. The highest RMSE



is for the least sensitive CMLs at site 7a (sensitivity 0.7 dB mm$^{-1}$ h$^{-1}$) and site 1a (sensitivity of 0.4 dB mm$^{-1}$ h$^{-1}$). Sites 2a and 6a, with the same sensitivity as 1a, reach, inconsistently, RMSE 0.6 and 0.8 mm h$^{-1}$.

The standard deviation of the binned rainfall intensities (by 2 mm h$^{-1}$) is in the range of 0.7 and 1.7 mm h$^{-1}$ for rainfall intensities under 10 mm h$^{-1}$, whereas for rainfall intensities between 10 and 20 mm h$^{-1}$ it increases slightly to a range of 1.0

and 3.4 mm h$^{-1}$. For all heavy rainfall intensities over 20 mm h$^{-1}$ it falls in the range of 1.0 to 4.9 mm h$^{-1}$. The median in the bins follows the direct proportionality well up to 50 mm h$^{-1}$.

To explore the systematic component of the measurement deviations, double-mass curves of cumulative rain of the CML pairs are displayed (Fig. 7). The lines show the main direction of the curves parallel to the diagonal, which indicates synchronized systematic errors of the independent sensors. However, changes in the trend of systematic errors can be observed

in the course of time. For example, by the pair with highest cumulative rainfall, link 2 observes systematically lower rain rates than link 1 up to rainfall depths around 200 mm, however, this trend then changes resulting in very low relative error between overall cumulative rainfalls at the end of the observation period. Overall, the pairs have a relative error between 0.01 and 0.18.

## 4.2 Evaluation of CML pairs operating at different frequency bands

Six sites contain pairs of CMLs operating in dissimilar frequency bands (indicated by "b" in the site name, see Fig. 1 and

Fig. 2). At sites 2b, 11b, and 12b the pairs combine 22 and 25 GHz frequency band CMLs. Site 3b combines 32 and 38 GHz frequency band CMLs while sites 4b and 5b combine 25 and 32 GHz frequency band CMLs.

Figure 8 shows a high correlation of 0.98 and low RMSE of 1.0 mm h$^{-1}$, on average, for all pairs of CMLs. Within the six pairs, the correlation ranges between 0.96 and 0.99 and RMSE between 0.6 and 1.7 mm h$^{-1}$. Sites 4b, 5b, and 12b indicate slightly higher RMSE (1.7, 1.1 and 1.2 mm h$^{-1}$). We did not find any clear link between CML characteristics (length,

frequency) and their magnitude of RMSE or other performance metrics. Overall, the performance worsened compared to the pairs operating at an identical frequency band.

The double-mass curves of cumulative rain of the CML pairs are also displayed for "b" sites (Fig. 9). All missing timesteps for one of the links in the pair were excluded. The lines show the main direction of the curves parallel to the diagonal, which indicates agreement of the independent sensors. However, CMLs operating at higher frequency bands (link 2) tend to

overestimate a CML operating at a lower frequency band (link 1). The worst agreement between CML pairs is observed at site 4b, where 32 GHz CML overestimates, compared to the 25 GHz CML, more than others with a relative error of 0.69. However, there are periods with direct proportionality behaviour, i.e., the curve is parallel to the line indicating perfect fit. An identical setup (32 and 25 GHz CMLs) at site 5b, with the same differences in frequency bands as 4b, does not mirror such a strong tendency for overestimation having a relative error of only 0.12. Furthermore, there are also a short steep rises in the course of

three other double-mass curves (sites 2b, 5b, and 11b). Two increases are associated with the longest stratiform event at sites 2b and 5b and are caused by systematic underestimation of rainfall intensity by links 1 in the pairs, while links 2 perceive low rainfall intensities (up to 4 mm h$^{-1}$) for a long time (approximately 11 hours of the event). Third steep rise is associated with a convective event at site 11b. However, in that event, a similar pattern occurred, the link 2 perceives low rainfall intensity





(around 2 mm h$^{-1}$) for approximately 4 hours between two rainfall peaks of the convective event. Overall, the pairs have a
relative error between 0.12 and 0.26 excluding site 4b. The relative error is, thus, higher than in the case of sites with identical
frequency CMLs in the pair.

A different frequency could affect wet antenna attenuation. Leinse et al. (2008) presented a ratio of rain-induced
attenuation to wet antenna attenuation where the dominance of the numerator to the denominator changes with respect to the
frequency. The CMLs at "b" sites are separated by 3 GHz (25 and 22 GHz), 6 GHz (38 and 32 GHz) and 7 GHz (32 and
25 GHz), however, the WAA model used a single set of parameters for all CMLs.

Berne and Uijlenhoet (2007) demonstrated the effects of DSD, which might be significant for inadequate coefficients
in power-law relation. They showed that DSD-related errors are dependent on CML frequency. This could partly explain lower
agreement between CML pairs of different frequencies observed in this study.

### 4.3 Effect of the type of event and time aggregation

Both convective and stratiform rainfalls (Fig. 10) have a similar quality of the match measured by the correlation coefficient
(0.92 - 0.98). RMSE is in the range 0.5 to 1.4 mm h$^{-1}$ for all combinations of rainfall and site types. RMSE decreases from
1.1 mm h$^{-1}$ to 0.6 mm h$^{-1}$ for stratiform events, as the rainfall intensity is lower for such events.

RMSE and correlation were calculated for time aggregations of 1, 5, 10, 15 and 30 min (Fig. 11) for both rainfall
types (convective and stratiform) and for both types of sites (sites "a" including the sites with identical frequency band CMLs,
and sites "b" with CMLs with different frequency bands). The aggregation brings a reduction of noisiness. The correlation
coefficient is consistently greater than 0.92. The increase in correlation is more pronounced for the time aggregation between
1 and 5 min. Further aggregation did not bring any additional improvement in this performance measure. The agreement of
devices expressed as RMSE improves particularly quickly for convective rainfall types in aggregation between 1 and 10 min
from RMSE 0.9 mm h$^{-1}$ to 0.6 mm h$^{-1}$ for pairs of identical frequency bands, and from 1.3 mm h$^{-1}$ to 1.1 mm h$^{-1}$ for pairs of
different frequency bands. Stratiform rainfall types do not rapidly improve in RMSE with greater time aggregation, but the
enhancement occurs linearly with greater time aggregation. On the other hand, for convective rainfall types improvement
accelerates, and further, it is even more pronounced for CML pairs at an identical frequency band.

### 4.4 Different processing methods

To demonstrate the effect of different separations of baseline and WAA, four CML processing methods were tested on a
smaller data set of a rainy month with cumulative rainfall of 76 mm in September 2014. The methodology of the processing
methods is described in Section 3.3.

As the complexity of processing methods increases, performance metrics improve. One needs to be aware that the
more a method reduces observed attenuation, the more zeros are in the obtained data, which would inadequately improve
performance metrics. The performance comparing the collocated measurements of the methods is summarized in Table 1.



Setting the subset criteria to light rainfall ($< 5$ mm h$^{-1}$) timesteps for at least one CML of the pair, method 0 is as good as, or outperforms, the other methods.

The results of this case study indicate that model 0 featuring the WAA separation method proposed by Pastorek et al. (2022) fits the data the best. Therefore, it was used as the CML rainfall retrieval method for the analysis presented in Sections 3.1 – 3.3.

## 5 Discussion

This section discusses the effects of different antenna hardware and evaluates the size of the dataset and data availability.

### 5.1 Effect of different hardware

There were three sites of collocated sensors operating at identical frequency bands but having a different hardware setup. The differences at the three sites are as follows. The antenna radome at one node at site 1a (25 GHz CMLs) has a diameter of 0.6 m,
unlike the 0.3 m diameters of the three remaining nodes. The antenna radome at one node at site 7a (38 GHz CMLs) is a Bosch Marconi PS UHP antenna, contrary to the MINILINK platform at the rest of the nodes. Each CML in the pair at site 10a (38 GHz CMLs) has one node with an antenna radome diameter of 0.6 m, unlike the 0.3 m diameter of the other nodes.

Despite the differences in hardware, CML pairs at these sites do not exhibit any differences in performance for rain-induced attenuation compared to the sites with an identical hardware setup. RMSE is in the range of 0.3 to 0.5 dB and the
correlation is between 0.96 and 0.98, which is in line with the other "a" sites discussed in Section 4.1.

The antenna radomes at site 12b are 0.6 m in diameter, which is the only distinction in antenna size or type within the metadata of the "b" sites. It makes this site, together with the almost twice as long CML paths at other sites, unique in the dataset. Compared to the other 25/22 GHz sites, the results at 12b are not as good with RMSE of 1.2 mm h$^{-1}$ and a correlation of 0.96 (sites 2b and 11b have RMSE 0.6 and 0.9 mm h$^{-1}$ and the correlation for both is 0.98).

Overall, the differences in hardware did not cause a significant decline in the performance of independent sensors. It needs to be noted, though, that the dataset contains only three sites of CMLs operating at identical frequency bands with different hardware and each of the hardware differences is unique.

### 5.2 Size of the dataset

The size of the dataset is affected by the data availability of both independent sensors and the visual inspection of erratic
behaviour. Therefore, not all sites are equally represented in the dataset. Data availability is summarized in Appendix A. Overall, there was higher availability for the "b" sites of CMLs operating at different frequency bands with a mean availability of 80 % (57 % – 100 %). The availability for "a" sites ranges between 46 % and 100 % with a mean of 66 %.

The lower number of values for the "a" sites is particularly striking in plots of the double-mass curves (Fig. 7 and 9), where cumulative rainfall reaches, on average, 190 mm in contrast to the "b" sites, which reach, on average, 248 mm.





370       The data availability for convective events is in the range of 52 % to 100 % (with a mean of 75 %) and for stratiform events 40 to 100 % (with a mean of 69 %). Subsetting the sites independently, the mean difference in availability of convective and stratiform event types is 6 % (with a range from -24 % to 23 %). Thus, convective events are slightly better represented across all the sites. It is worth mentioning that the duration of the periods of both event types selected for the study are similar (153 h for convective events and 151 h and 15 min for stratiform events).

## 6 Conclusions

The presented study investigates the consistency between different CMLs by comparing pairs of independent collocated devices operating at the same and different frequencies, including the identification of random and systematic components of CML discrepancies. The study empirically evaluates errors of CML observations without reference rainfall measurements which enables assessment of hardware homogeneity. In total, eight pairs of independent collocated K-band CMLs operating at identical frequency bands and six pairs operating at different frequency bands were analysed in the study at high temporal resolution (1 min).

      Rain-induced attenuation of collocated CMLs operating at identical frequency bands shows very good agreement. The RMSE in rain-induced attenuation is 0.4 dB, which is close to signal quantization level. The random error is low in total CML observation error.

      For collocated CMLs operating at identical frequency bands, RMSE (differences) of the retrieved rainfall information are between 0.4 and 0.9 mm h$^{-1}$ and have correlations between 0.97 and 0.99. Collocated CMLs operating at different frequency bands perform less consistently with RMSE between 0.6 and 1.7 mm h$^{-1}$. The correlation has similar values between 0.96 and 0.99. Additionally, separating the rainfall types, the correlation is higher than 0.92 for both convective and stratiform events. The RMSE is lower for stratiform events as such events have overall lower rainfall intensities. The systematic errors are more pronounced for collocated CMLs operating at different frequency bands. CMLs operated at higher frequency tend to observe higher rain rates than those operated at lower frequencies. Overall, the observations are in very good agreement, which confirms homogeneous behaviour of the hardware in the real network.

      It is shown that the more complex the processing method, the better is the agreement between collocated CMLs. This also applies to rainfalls with lower intensity. This study also provides some insight into collocated CMLs of different radome sizes and antenna types. It might be interesting to evaluate collocated CMLs of different antenna types and sizes within a real CML network if a greater dataset was available in further studies.

      Even though CMLs in real cellular communication networks have not been, first and foremost, deployed for rainfall monitoring, the independent collocated CML sensors are in excellent agreement and the hardware is homogeneous in its behaviour. The performance is at a similar level as for collocated rain gauges (e.g., Peleg et al. (2013) demonstrated rain gauge correlation 0.92 for 1 min resolution).



The collocated CMLs operating at identical frequency bands perform consistently. This confirms that the data of CMLs of the same characteristics can be processed with identical parameters for rainfall retrieval models.

*Competing interests.* The authors declare that they have no conflict of interest.

*Acknowledgements.* The authors greatly acknowledge financial support from the Czech Science Foundation (GACR) project SpraiLINK 20-14151J and the Czech Technical University in Prague project no. SGS22/045/OHK1/1T/11. We would like to
thank T-Mobile Czech Republic a.s. for providing the CML data. Special thanks are extended to Pražská vodohospodářská společnost a.s. for providing rainfall data from their rain-gauge network and Pražské vodovody a kanalizace, a.s., who carefully maintains the rain gauges. Last but not least, we would like to thank Christian Chwala and Nico Blettner for introducing alternative CML processing and radar adjustment methods.

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



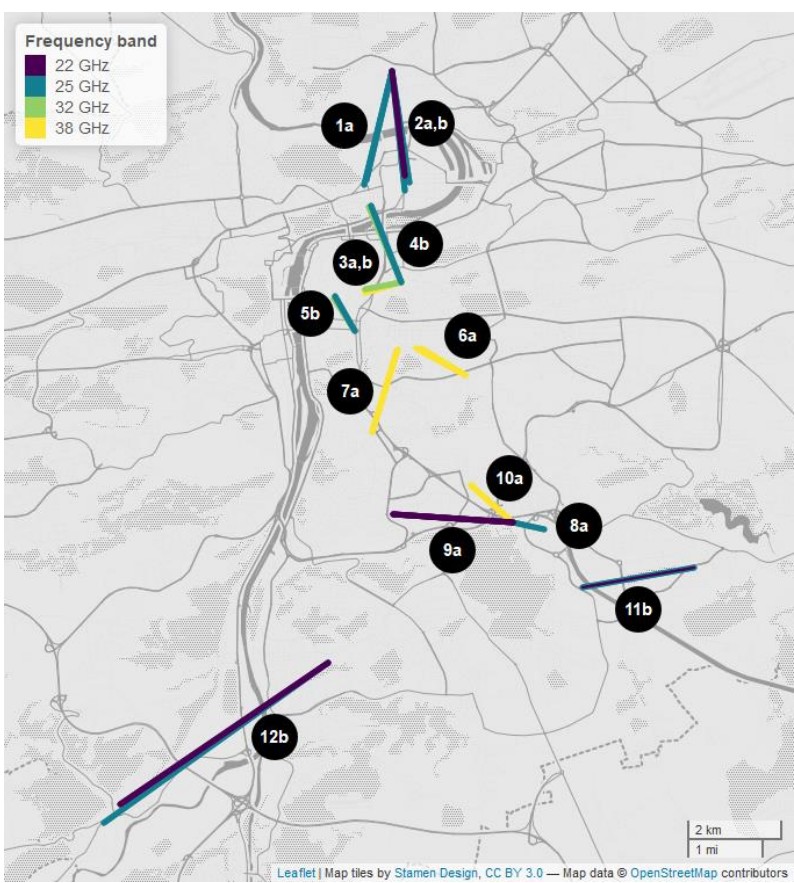

**Figure 1: Map of the 12 sites of collocated CMLs in the Prague CML network of the provider T-Mobile. The line colour indicates the frequency band at which CMLs operate. The "a" in the name of the site indicates pairs of CMLs having identical frequency bands and "b" indicates pairs of CMLs having different frequency bands.**






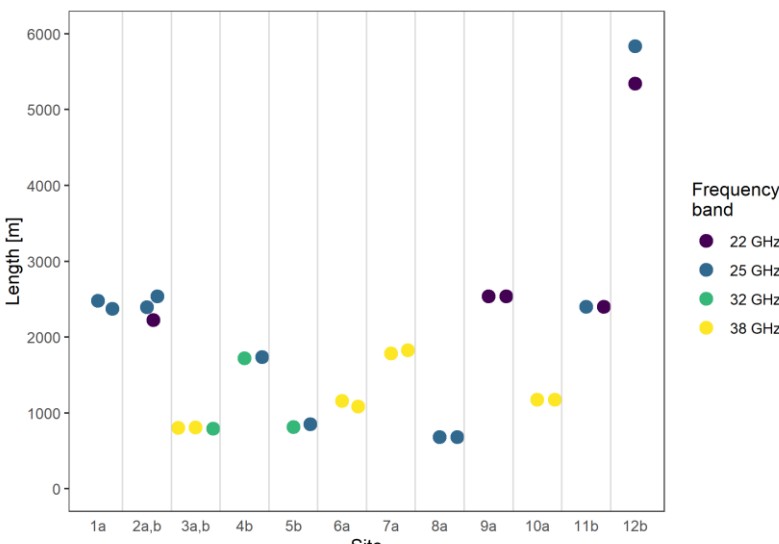

**Figure 2: Distribution of lengths and frequencies of CMLs used in this study. The colours indicate the frequency band at which CMLs operate. The "a" in the name of the site indicates pairs of CMLs having identical frequency bands and "b" indicates pairs of CMLs having different frequency bands.**

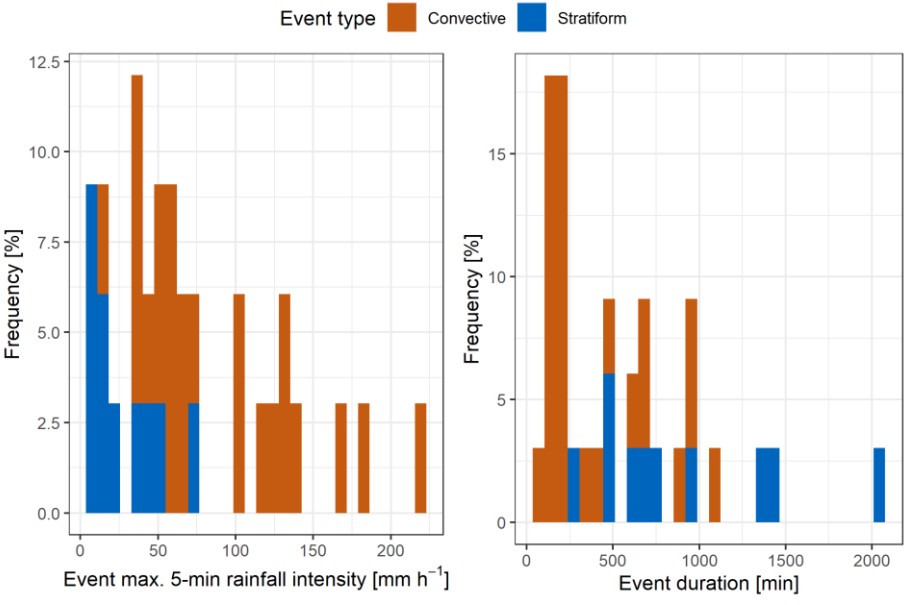

**Figure 3: Characteristics of 33 selected rainfall events. (left) Histogram of maximum event rainfall intensity based on 5-min moving average of 23 municipal rain gauge measurements. (right) Histogram of event durations. The colour indicates the type of rainfall event (convective or stratiform)**





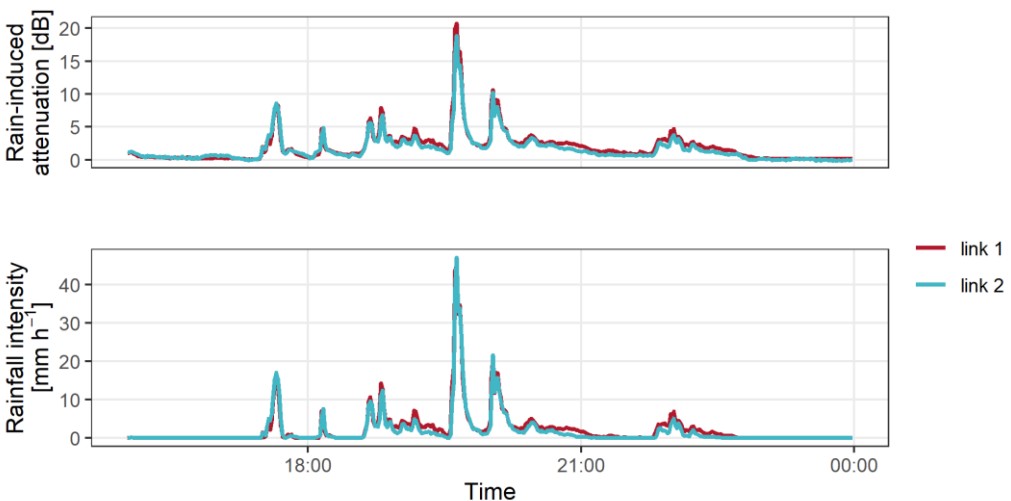

**Figure 4: Example of a time series of collocated independent sensors in group 2a for (top) rain-induced attenuation and (bottom) rainfall intensity on 21st July 2014.**

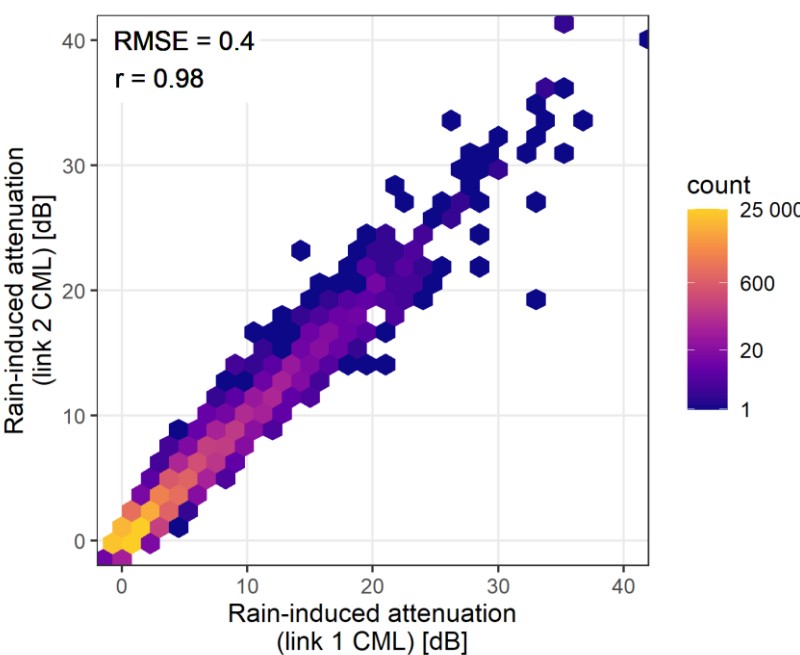

**Figure 5: Scatter density plot of rain-induced attenuation for 8 collocated CML pairs having identical frequencies. The axis range of the plot is cut to 40 dB as there were only 4 data points out of this range. The performance metrics (correlation and RMSE in dB) are in the top left corner.**





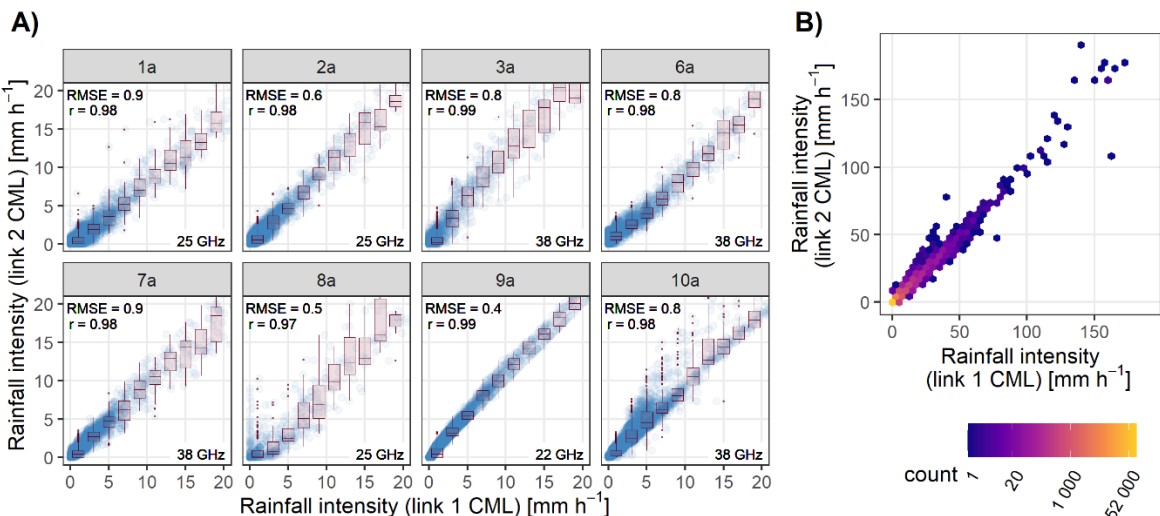

**Figure 6: (left) Scatter plot of CML-derived rainfall intensity for 5 collocated CML pairs having identical frequencies. Values under 20 mm h⁻¹ are shown. The boxplots show a spread when binned to 2 mm h⁻¹ bins by link 1. The performance metrics (correlation and RMSE in mm h⁻¹) are in the top left corner and the frequency band is indicated in the bottom right corner for each site (right) Scatter density plot of rainfall intensity for all pairs at the full range of values.**

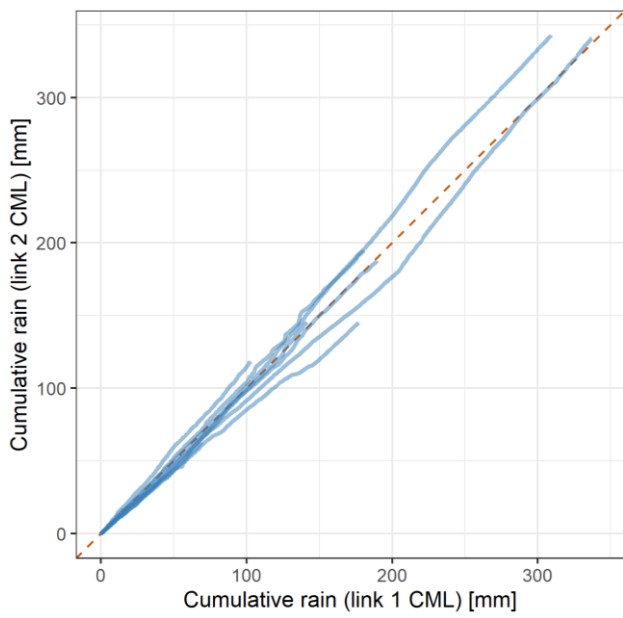

**Figure 7: Double-mass curve for collocated sensors operating at identical frequency bands.**





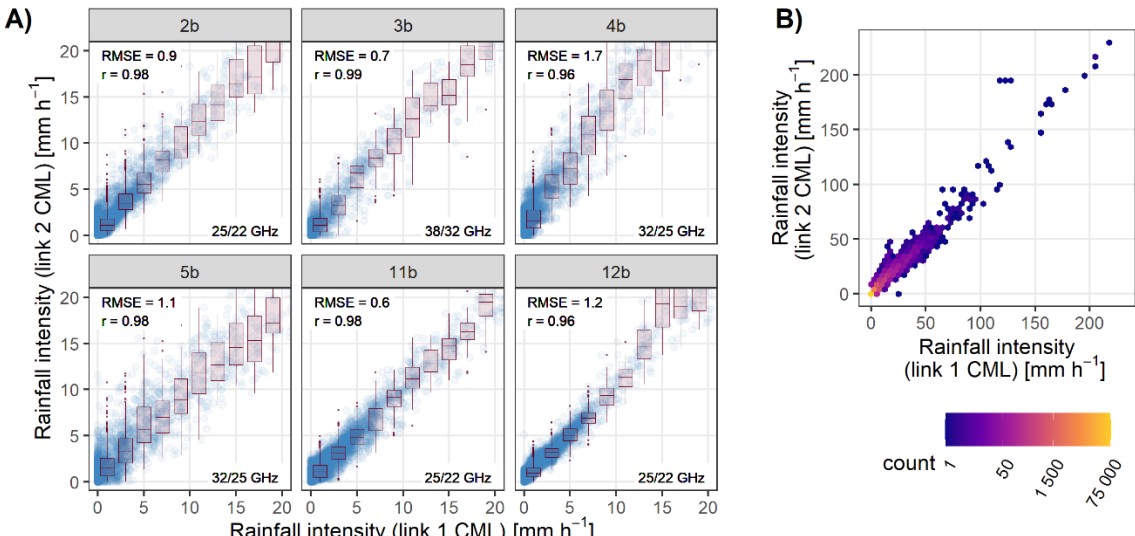

**Figure 8: (left)** Scatter plot of CML-derived rainfall intensity for 6 collocated CML pairs having different frequencies. Values under 20 mm h$^{-1}$ are shown. The boxplots show a spread when binned to 2 mm h$^{-1}$ bins by link 1. The performance metrics (correlation and RMSE in mm h$^{-1}$) are in the top left corner and the frequency band is indicated in the bottom right corner for each site **(right)** Scatter density plot of rainfall intensity for all pairs at the full range of values.

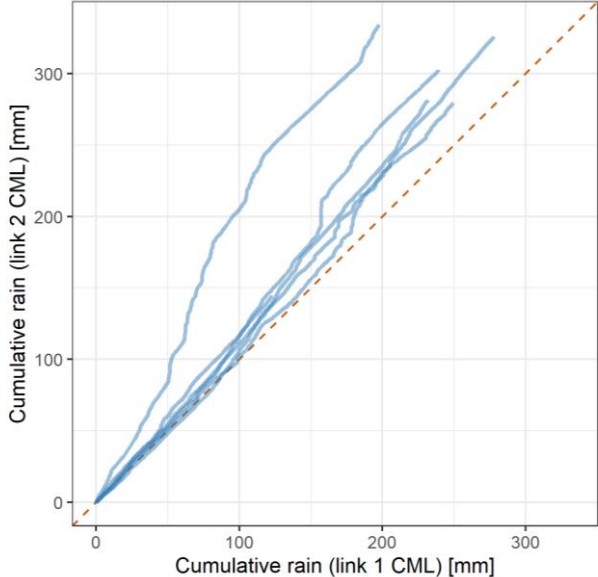

**Figure 9:** Double-mass curve for collocated sensors operating at different frequency bands. The curve, which differs the most from the line $y = x$, belongs to site 4b.



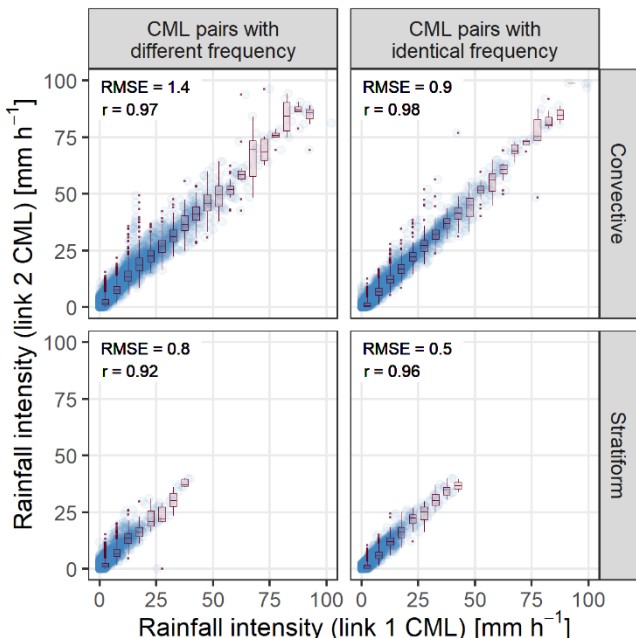

**Figure 10: Scatter plots of CML-derived rainfall intensity for collocated CML pairs of (left column) different frequency band and (right column) identical frequency band in 1-min temporal resolution for (top row) convective and (bottom row) stratiform rainfall events. In the top left corner are the performance measures (correlation and RMSE in mm h⁻¹). Values under 100 mm h⁻¹ are shown.**

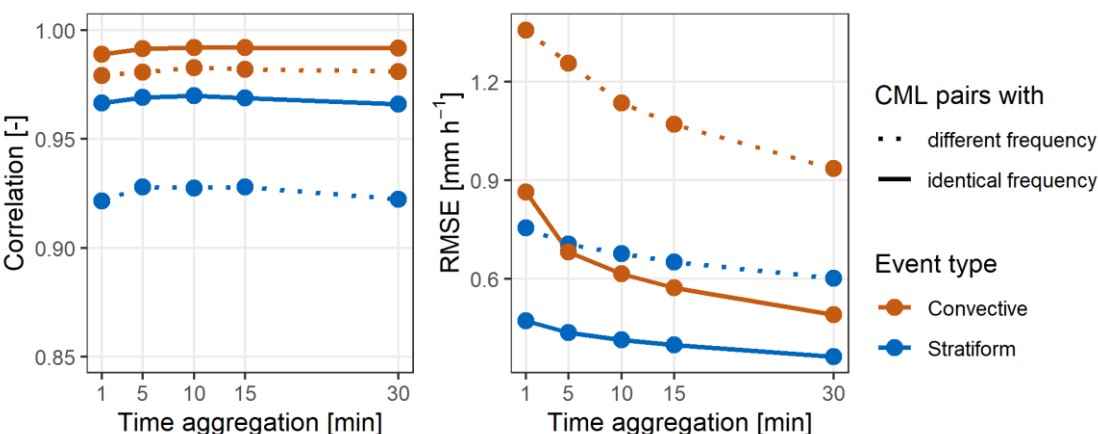

**Figure 11: Influence of time aggregation (1, 5, 10, 15 and 30 min) on (left) correlation and (right) RMSE in mm h⁻¹.**



**Table 1: Comparison of the performance of four processing methods.**

| Method | Threshold | Correlation (-) | RMSE (mm h$^{-1}$) |
|---|---|---|---|
| 0 | not applied | 0.97 | 0.7 |
|   | R < 5 mm h$^{-1}$ | 0.85 | 0.5 |
| 1 | not applied | 0.94 | 1,7 |
|   | R < 5 mm h$^{-1}$ | 0.71 | 1.4 |
| 2 | not applied | 0.95 | 0.9 |
|   | R < 5 mm h$^{-1}$ | 0.75 | 0.7 |
| 3 | not applied | 0.96 | 0.7 |
|   | R < 5 mm h$^{-1}$ | 0.74 | 0.5 |





**Appendix A: Metadata table of CMLs**

The following table summarises the characteristics of the CMLs at the sites. The sites indicated by "a" are operated at identical
frequency bands and those indicated by "b" are operated at different frequency bands. The frequency and parameters $\alpha$ and
$\beta$ are averaged for both sublinks of one link. The last column indicates the percentage of pairwise data availability relative to
the duration of all events.

**Table A1: Characteristics of the CMLs at the sites and pairwise data availability.**

| Site | Link | Frequency (GHz) | Polarization* | Path length (m) | $\alpha$ (mm h$^{-1}$ dB$^{-\beta}$ km$^{\beta}$); $\beta$ (-) | Antenna size node A (m) | Antenna size node B (m) | Antenna type node A | Antenna type node B | Pairwise availability (%) |
|---|---|---|---|---|---|---|---|---|---|---|
| 1a | 1 | 25.06 | H | 2374 | 7.195; 1.053 | 0.3 | 0.3 | UKY 21074/SC15 | UKY 21074/SC15 | 52 |
| 1a | 2 | 25.06 | V | 2480 | 6.364; 1.001 | 0.3 | 0.6 | UKY 21074/SC15 | UKY 22046/SC15 | 52 |
| 2a, b | 2 | 25.06 | V | 2538 | 7.195; 1.053 | 0.3 | 0.3 | UKY 21074/SC15 | UKY 21074/SC15 | 50 ("2a"); 84 ("2b") |
| 2a | 1 | 25.06 | H | 2393 | 6.364; 1.001 | 0.3 | 0.3 | UKY 22070/SC15 | UKY 22070/SC15 | 50 |
| 2b | 1 | 22.55 | H | 2225 | 7.736; 0.976 | 0.3 | 0.3 | UKY 21073/SC15 | UKY 21073/SC15 | 84 |
| 3a, b | 2 | 37.97 | H | 805 | 2.837; 1.130 | 0.3 | 0.3 | UKY 21075/SC15 | UKY 21075/SC15 | 50 ("3a"); 57 ("3b") |
| 3a | 1 | 38.25 | H | 804 | 2.797; 1.132 | 0.3 | 0.3 | UKY 21075/SC15 | UKY 21075/SC15 | 50 |
| 3b | 1 | 32.23 | H | 792 | 3.911; 1.081 | 0.3 | 0.3 | UKY 22072/SC15 | UKY 22072/SC15 | 57 |
| 4b | 1 | 25.06 | V | 1734 | 7.195; 1.053 | 0.3 | 0.3 | UKY 21074/SC15 | UKY 21074/SC15 | 87 |
| 4b | 2 | 32.63 | H | 1718 | 3.815; 1.086 | 0.3 | 0.3 | UKY 22072/SC15 | UKY 22072/SC15 | 87 |
| 5b | 1 | 25.06 | V | 851 | 7.195; 1.053 | 0.3 | 0.3 | UKY 21074/SC15 | UKY 21074/SC15 | 100 |
| 5b | 2 | 32.23 | V | 814 | 4.333; 1.112 | 0.3 | 0.3 | UKY 22072/SC15 | UKY 22072/SC15 | 100 |
| 6a | 1 | 38.25 | V | 1157 | 3.019; 1.172 | 0.3 | 0.3 | UKY 21075/SC15 | UKY 21075/SC15 | 50 |





| Site | Link | Frequency (GHz) | Polarization* | Path length (m) | $\alpha$ (mm h$^{-1}$ dB$^{-\beta}$ km$^{\beta}$); $\beta$ (-) | Antenna size node A (m) | Antenna size node B (m) | Antenna type node A | Antenna type node B | Pairwise availability (%) |
|---|---|---|---|---|---|---|---|---|---|---|
| 6a | 2 | 37.97 | V | 1085 | 3.069; 1.170 | 0.3 | 0.3 | UKY 22073/SC15 | UKY 22073/SC15 | 50 |
| 7a | 1 | 38.25 | V | 1785 | 3.019; 1.172 | 0.3 | 0.3 | UKY 21075/SC15 | UKY 21075/SC15 | 46 |
| 7a | 2 | 37.97 | V | 1825 | 3.069; 1.170 | 0.3 | 0.3 | UKY 21075/SC15 | PS 0.3-38.0/1P UHP | 46 |
| 8a | 1 | 25.06 | H | 682 | 6.364; 1.001 | 0.3 | 0.3 | UKY 21074/SC15 | UKY 21074/SC15 | 82 |
| 8a | 2 | 25.56 | H | 682 | 6.122; 1.006 | 0.3 | 0.3 | UKY 21074/SC15 | UKY 21074/SC15 | 82 |
| 9a | 1 | 22.51 | V | 2534 | 8.777; 1.035 | 0.3 | 0.3 | UKY 21073/SC15 | UKY 21073/SC15 | 100 |
| 9a | 2 | 22.51 | V | 2534 | 8.777; 1.035 | 0.3 | 0.3 | UKY 21073/SC15 | UKY 21073/SC15 | 100 |
| 10a | 1 | 37.90 | V | 1171 | 3.078; 1.169 | 0.3 | 0.6 | UKY 21075/SC15 | UKY 21080/SC15 | 100 |
| 10a | 2 | 38.53 | V | 1171 | 2.972; 1.176 | 0.3 | 0.6 | UKY 21075/SC15 | UKY 21080/SC15 | 100 |
| 11b | 1 | 22.51 | V | 2397 | 8.777; 1.035 | 0.3 | 0.3 | UKY 21073/SC15 | UKY 21073/SC15 | 94 |
| 11b | 2 | 25.06 | V | 2397 | 7.195; 1.053 | 0.3 | 0.3 | UKY 21074/SC11 | UKY 21074/SC11 | 94 |
| 12b | 1 | 22.51 | V | 5341 | 8.777; 1.035 | 0.6 | 0.6 | UKY 21078/SC15 | UKY 21078/SC15 | 57 |
| 12b | 2 | 25.06 | V | 5836 | 7.195; 1.053 | 0.6 | 0.6 | UKY 21079/SC15 | UKY 21079/SC15 | 57 |

\* "V" means vertical, and "H" means horizontal




## Appendix B: Overview of rainfall events

**Table B1: Rainfall events selected for the analysis.**

| Event no. | Event start (UTC) | Event end (UTC) | Duration (min) | Max. 5-min rainfall intensity (mm h$^{-1}$) | Event type |
|---|---|---|---|---|---|
| 1 | 08.07.2014 9:40 | 09.07.2014 3:30 | 1070 | 135 | Convective |
| 2 | 11.07.2014 13:20 | 11.07.2014 15:45 | 145 | 121 | Convective |
| 3 | 13.07.2014 13:00 | 13.07.2014 15:00 | 120 | 58 | Convective |
| 4 | 21.07.2014 12:55 | 22.07.2014 0:30 | 695 | 165 | Convective |
| 5 | 28.07.2014 13:15 | 28.07.2014 15:10 | 115 | 54 | Convective |
| 6 | 30.07.2014 14:50 | 30.07.2014 18:10 | 200 | 218 | Convective |
| 7 | 11.08.2014 0:00 | 11.08.2014 12:05 | 725 | 18 | Stratiform |
| 8 | 16.08.2014 4:25 | 16.08.2014 20:05 | 940 | 36 | Convective |
| 9 | 26.08.2014 3:40 | 26.08.2014 14:10 | 630 | 6 | Stratiform |
| 10 | 26.08.2014 20:15 | 27.08.2014 7:10 | 655 | 7 | Stratiform |
| 11 | 08.09.2014 14:40 | 08.09.2014 21:20 | 400 | 140 | Convective |
| 12 | 11.09.2014 14:55 | 12.09.2014 13:20 | 1345 | 54 | Stratiform |
| 13 | 14.09.2014 2:30 | 14.09.2014 18:30 | 960 | 47 | Convective |
| 14 | 19.09.2014 20:05 | 19.09.2014 21:55 | 110 | 68 | Convective |
| 15 | 21.09.2014 19:00 | 21.09.2014 23:55 | 295 | 39 | Stratiform |
| 16 | 13.10.2014 22:00 | 14.10.2014 6:05 | 485 | 25 | Stratiform |
| 17 | 16.10.2014 4:20 | 16.10.2014 7:40 | 200 | 17 | Convective |
| 18 | 16.10.2014 15:45 | 16.10.2014 16:40 | 55 | 68 | Convective |
| 19 | 27.04.2015 18:25 | 27.04.2015 21:35 | 190 | 73 | Convective |
| 20 | 29.05.2015 20:50 | 29.05.2015 23:55 | 185 | 99 | Convective |
| 21 | 08.06.2015 18:15 | 09.06.2015 8:55 | 880 | 58 | Convective |
| 22 | 27.07.2015 11:25 | 27.07.2015 15:15 | 230 | 58 | Convective |





| Event no. | Event start (UTC) | Event end (UTC) | Duration (min) | Max. 5-min rainfall intensity (mm h⁻¹) | Event type |
|---|---|---|---|---|---|
| 23 | 17.08.2015 4:40 | 18.08.2015 14:30 | 2030 | 43 | Stratiform |
| 24 | 04.05.2016 2:10 | 04.05.2016 18:35 | 985 | 14 | Stratiform |
| 25 | 24.05.2016 13:25 | 24.05.2016 23:35 | 610 | 52 | Convective |
| 26 | 31.05.2016 3:20 | 31.05.2016 6:10 | 170 | 35 | Convective |
| 27 | 02.06.2016 22:00 | 03.06.2016 6:20 | 500 | 7 | Stratiform |
| 28 | 12.06.2016 12:15 | 12.06.2016 17:55 | 340 | 103 | Convective |
| 29 | 17.06.2016 3:05 | 17.06.2016 11:35 | 510 | 35 | Convective |
| 30 | 01.07.2016 0:20 | 01.07.2016 2:20 | 120 | 115 | Convective |
| 31 | 13.07.2016 15:05 | 14.07.2016 14:50 | 1425 | 75 | Stratiform |
| 32 | 26.07.2016 12:50 | 26.07.2016 16:40 | 230 | 179 | Convective |
| 33 | 16.09.2016 18:20 | 17.09.2016 6:05 | 705 | 132 | Convective |
