# Peer review of "Evaluation of error components in rainfall retrieval from collocated commercial microwave links"

_Atmospheric Measurement Techniques, 2022_

## Referee Comment (RC2)

**Evaluation of error components in rainfall retrieval from collocated commercial microwave links**

*Anna Špačková, Martin Fencl, Vojtěch Bareš*

The authors present yet another work on commercial microwave links (CMLs), widening the scope and understanding of such an interesting topic, which has gained momentum during the last two decades. The aim of this paper is to show the performance of collocated CMLs, i.e., CMLs sharing the same link-path, and receiving and transmitting antennas (operating at different frequencies though). Although past studies have dealt too with collocated CMLs, the approach that the authors present in this work is innovative as they expand, and mainly focus, on the performance of collocated CMLs against not collocated ones.

The manuscript needs some re-working as it is not perfectly written, something which at some point interferes with the narrative of the idea/message. Hence, I suggest its publication after some (quasi-) major changes.

**MAJOR COMMENTS**

– pag. 7, line 197: What the maximum 5-min intensity is computed for?. There seems to be no use of such estimates further down the analyses. Besides, further ahead in line 198, the median of the maximum 5-min intensities is not shown in Figure 3 (as the authors seem to suggest).

– pag. 8, lines 219 to 227: Enclose these two paragraphs into a section titled 'Radar data', for instance.

– pag. 8, line 231; and pag. 9, line 246: Equations (8) and (9) are very basic equations that can be removed, given that they are not essential in the analyses nor in the results presented in this manuscript.

– pag. 8, lines 232 to 241; and pag. 11-12, lines 333 to 341, pag. 24, Table 1: The authors propose 3 alternative methods to estimate baseline and WAA. Nevertheless, the description of such methods is not entirely clear. Furthermore, the authors do not offer a solid argument (neither do they present compelling results/figures) on how these alternatives would benefit potential future studies (which apparently is not the case, given what they conclude in pag. 12, lines 342 to 344). Overall, this unclarity and lack of results leads to confusion, and aimlessness with regard to alternatives in baseline and WWA estimates. What I suggest is that the authors either offer a more in-depth view of the alternatives they present, and the benefits in implementing them (with results included); or do not mention/present at all such alternatives, thus avoiding obscuring the simple/basic idea they center their work around. If the authors proceed with presenting these 3 alternatives, the enumeration (of the alternatives) should be 1 to 4, instead of 0 to 3 (as it presently is). It also worth mentioning in your description/analyses of Wet/Dry classification periods, works such as Song et al. [1], for instance.

– pag. 11, line 323: Instead of analyses for $10 - \min$ aggregations, I'd prefer to see aggregations at $1 - h$. In practical applications $5$ and $15 - \min$ aggregations are near enough (to $10 - \min$). Nevertheless, having hourly aggregates/estimates would allow comparisons with other rainfall products (e.g., satellite).

– pag. 19, Figure 2: This would be a more valuable plot if it were presented as a scatterplot of Frequency vs. Length (in the style of Figure 2, of your reference in line 420; or Figure 7, of your reference in line 428; or Figure 2, of your reference in line 443); which it is more common and "standard" in CML studies. If any color scale is to be applied, that could be site-code, for instance.

– pag. 20, Figure 5; and pag. 21, Figure 6B; and pag. 22, Figure 8B: Join these figures into one (i.e., one figure three panels).

– pag. 21, Figure 6A; and pag. 22, Figure 8A: Join these figures into one (i.e., one figure two panels).

– pag. 21, Figure 7; and pag. 22, Figure 9: Join these figures into one (i.e., one figure two panels).

**MINOR COMMENTS**

– pag. 1, line 10: Replace 'reference' by 'reference to'.

- pag. 1, line 13: Replace '2014 and 2016' by '2014 - 2016' (given that your study goes from 2014 through 2016, right?).

- pag. 1, line 13: Replace 'in 1-min' by 'at 1-min'.

- pag. 1, lines 14, and 15: Replace 'commercial microwave links' by 'CMLs' (from line 1 the authors established the acronym, so why not use it here too?).

- pag. 1, line 27: Replace 'studies' by 'studies have'.

- pag. 2, line 27: Replace 'have not been, to the best of our knowledge, reported' by 'have not been reported, to the best of our knowledge'.

- pag. 3, line 64: Replace 'or' by 'and'.

- pag. 3, line 68: Replace 'the same frequencies and at' by 'equal and'.

- pag. 3, line 69: Replace 'these' by 'such'.

- pag. 3, line 75: Remove 'finally,'.

- pag. 3, line 77: Replace 'utilizes' by 'uses'.

- pag. 3, line 78: Replace 'total loss' by 'total path loss'.

- pag. 3, line 79: Replace 'rainy' by 'wet'.

- pag. 3, line 79: Replace 'and accounting for' by 'plus the'.

- pag. 3, line 79: Replace 'that is caused by water' by 'caused by the water'.

- pag. 3, line 80: Replace 'forming on antenna' by 'accrued on the antenna'.

- pag. 3, line 80: Replace 'rain and dew' by 'rain and/or dew'.

- pag. 3, line 80: Add your reference 'Leth et al., 2018' (which should actually be 'van Leth et al., 2018') after/before 'Chwala and Kunstmann, 2019' (please, note that is 'Chwala', and not 'Chawla'). Double check all your references for such inconsistencies.

- pag. 3, line 80: 'The specific raindrop attenuation...'. You have so far not defined what 'specific attenuation' is. Please, add accordingly something like: ', which is the total path loss divided by the link-path distance...'.

- pag. 3, line 81: Replace 'estimation' by 'estimates'.

- pag. 3, line 91: Replace '0.3 resp. 1 dB' by '0.3 and 1 dB, respectively'. Furthermore, the authors seem to use (quite often) 'resp.' as an acronym of 'respectively', something I have personally never seen before. Therefore, please change accordingly all the 'resp.' instances (e.g., lines 97, 186, 200).

- pag. 3, line 92: Please specify again what the 'second case' refers to.

- pag. 4, lines 98, and 100: Remove 'losses due to'.

- pag. 4, line 99: Replace 'losses causing' by 'caused by the'.

- pag. 4, line 100: Remove 'wet antenna attenuation' (The authors already established the acronym in pag. 3).

- pag. 4, lines 101, and 102: Remove this sentence. The authors already established what WAA is (

- pag. 3, line 80).

- pag. 4, line 103: Replace 'separated' by 'regard to'.

- pag. 4, line 106: Remove ', thus,'.

- pag. 4, line 108: Replace 'necessarily fulfilled' by 'entirely accurate'.

- pag. 4, line 109: Replace ', as well as temperature, can' by 'and temperature'.

- pag. 4, line 110: I'd suggest replacing 'raindrop path attenuation' by 'attenuation from raindrops'; in all possible instances, e.g., lines 131, and 134.

- pag. 4, line 111: Replace 'can' by 'do'.

- pag. 4, line 119: Replace 'WAA magnitude and pattern' by 'patterns of WAA magnitude'.

- pag. 4, line 121: Replace 'a power' by 'the power'.

- pag. 5, line 124: Replace 'on CML' by 'on the CML'.

- pag. 5, line 125: Use the DSD acronym first in pag. 2, line 36 (not here).

- pag. 5, line 125: Replace 'alpha and beta' by '$\alpha$ and $\beta$'.

- pag. 5, line 128: Replace 'are lowest' by 'are the lowest'.

- pag. 5, line 130: Replace 'of total' by 'of the total'.

- pag. 5, line 131: Replace 'by CML' by 'by the CML'.

- pag. 5, line 132: Replace 'incorrectly estimated' by 'wrong estimates of'.

- pag. 5, line 133: Replace '$k - R$ relation non-linearity' by 'the non-linearity of the $k - R$ relation'.

- pag. 5, line 137: Replace 'The same collective of authors' by 'Fencl et al., 2020 also' (and remove this reference at the end of the sentence).

- pag. 5, line 139: Remove 'alone'.

- pag. 5, line 142: Remove 'can, however,'.

- pag. 5, line 143: Remove 'Losses on the antennas also include WAA caused by the formation of a water layer on the antenna radomes.' (too repetitive by now).

- pag. 5, line 147: Replace 'EM' by 'Eletromagnetic (EM)' (this acronym was not previously established).

- pag. 5, line 148: Replace ', however, they interact' by 'interacting'.

- pag. 5, line 148: Replace 'Discrepancies' by 'Therefore, discrepancies'.

- pag. 5, line 149: Remove 'thus'.

- pag. 5, line 150: Replace 'the same frequency' by 'equal'.

- pag. 5, lines 152 to 154: Remove the last sentence (it's repetitive of the above sentence, and adds confusion instead of clarity).

- pag. 6, line 156: Replace 'twelve' by '12'.

- pag. 6, line 160: Remove 'one pair of' (in both instances).

- pag. 6, line 161: Replace 'intercompared for' by 'compared among'.

- pag. 6, line 151: Replace 'frequency bands' by 'frequencies' (here and everywhere else in the manuscript; e.g., lines 169, 244, 247, 263, 264, 288, 289, 382, 385, 386, 515, 570).

- pag. 6, line 162: Replace 'sites' by 'the sites'.

- pag. 6, line 164: Remove the title 'Materials'.

- pag. 6, line 165: Enumerate 'CML' as 3.1.

- pag. 6, line 168: Replace 'To compare, such' by 'Such'.

- pag. 6, line 169: Replace 'urbanized' by 'urban'.

- pag. 6, line 170: Replace '682 m to 5836 m' by '$\sim 0.7$ to $\sim 5.8\,\mathrm{km}$'.

- pag. 6, line 173: Replace 'having' by 'having a'.

– pag. 6, line 173: Replace 'are 0.3 m or 0.6 m in diameter' by 'have diameter of $30$ or $60\,\mathrm{cm}$'.

– pag. 6, line 174: Replace 'polarizations, are' by 'polarization is'.

– pag. 6, line 176: Replace 'respectively' by ', respectively'.

– pag. 6, line 177: Enumerate 'Gauge-adjusted radar rainfall' as 3.2 (and re-enumerate the remaining titles accordingly).

– pag. 6, line 180: Replace 'permanent municipal RG' by 'municipal rain gauge' (as the authors do not use anywhere else the 'RG' acronym).

– pag. 6, line 182: Replace 'monthly and regularly dynamically calibrated' by 'and regularly calibrated on a monthly basis'.

– pag. 6, line 186: Replace 'spatial and temporal' by 'spatiotemporal'.

– pag. 6, line 186: Replace 'resp.' by 'every'.

– pag. 7, line 188: Replace 'uses a data set collected during the monitoring of the' by 'is for the monitoring'.

– pag. 7, line 188: Replace '2014 and' by '2014 through'.

– pag. 7, line 189: Replace 'In total' by 'In total,'.

– pag. 7, line 190: Replace 'one hour' by '$1\,\mathrm{h}$'.

– pag. 7, line 191: Replace '304 h and 15 min' by '$\sim 304\,\mathrm{h}$'.

– pag. 7, line 191: Remove 'and 15 min'.

– pag. 7, line 194: Replace 'First' by 'one'.

– pag. 7, line 195: Replace 'Second' by 'the other'.

– pag. 7, line 196: Replace 'ten' by '10'.

– pag. 7, line 197: Replace 'maximal' by 'maximum'.

– pag. 7, line 198: Replace 'median maximal' by 'average maximum'.

– pag. 7, line 199: Replace 'reached up to' by 'was'.

– pag. 7, line 199: Replace 'events up to' by 'events was'.

– pag. 7, line 199: Replace 'median' by 'average'.

– pag. 7, line 202: Remove 'temporal'.

– pag. 7, line 203: It is not clear how the authors perform 1-min aggregations 'using averaging'.

– pag. 7, line 210: Replace 'wet antenna attenuation model was' by 'WAA model is'.

– pag. 7, line 213: Replace 'maximum $Aw$' by 'maximum WAA' (why not use WAA instead of Aw in Equation (6)?).

– pag. 8, line 216: Replace 'product grid cells' by 'pixels'.

– pag. 8, line 225: Replace 'The data were resampled to 1 h timesteps' by 'Radar data was aggregated from 5-min to 1-h'.

– pag. 8, line 230: Remove 'of the' from 'mean of the rainfall'.

– pag. 8, line 243: Remove '(Figure 2)'.

– pag. 9, line 250: Replace ':' by ', i.e., '.

– pag. 9, line 252: Replace 'which one' by 'which at least one'.

– pag. 9, line 253: Remove the first sentence.

- pag. 9, line 256: Please clarify what do you mean by 'the deviation' (deviation from what?).

- pag. 9, line 258: Remove the last sentence (of the paragraph).

- pag. 9, lines 261 to 262; and pag.12, line 346: Please, remove this "filling" text (no need for it).

- pag. 9, line 264: Remove 'are evaluated in this subsection'.

- pag. 9, line 265: Replace 'demonstrates' by 'shows'.

- pag. 9, line 265: Remove 'at two stages of processing: rain-induced attenuations are shown in the top panel, and rain rates in the bottom panel'.

- pag. 9, line 268: Remove ', which, on the other hand, is not evident from the scatterplots (Fig. 6)'.

- pag. 9, line 269: Replace 'represents all eight' by 'shows all 8'.

- pag. 9, line 272: Replace 'To conclude, this' by 'This'.

- pag. 9, line 273: Replace 'is low in total CML observation error' by 'in total CML observation error is low'.

- pag. 9, line 274: Replace 'Scatterplots' by 'Figure 6 shows scatterplots'.

- pag. 10, line 276: What is 'sensitivity'?; and how it's measured? (do you mean 'uncertainty', maybe?).

- pag. 10, line 283: Replace '(Fig. 7)' by 'in Figure 7'.

- pag. 11, line 310: Remove ', thus,'.

- pag. 11, line 315: When the authors say 'the WAA model used', to what WAA-model exactly you're referring to? (please see my "major" comment in regard to the models of WAA (and baseline) estimation).

- pag. 11, line 311: What do you mean by 'match'.

- pag. 11, line 332: Remove 'further,'.

- pag. 12, lines 358, and 359: It is not clear form this last sentence whether the antenna size does influence (or not) the performance the rainfall retrievals. Please clarify.

- pag. 13, line 376: Replace 'The presented' by 'This'.

- pag. 13, line 378: Replace 'The' by 'This'.

- pag. 13, line 385: Replace 'RMSE (differences) of the retrieved rainfall information' by 'differences in RMSE of rainfall retrievals'.

- pag. 13, line 386: Please update all instances of $\mathrm{mm\ h^{-1}}$ to $\mathrm{mm \cdot h^{-1}}$.

- pag. 13, line 388: Replace 'types,' by 'types (convective and stratiform),'.

- pag. 13, line 393: Remove this first sentence (please also see my "major" comment in regard to the models of WAA (and baseline) estimation).

- pag. 11, line 397: Remove ', first and foremost,'. This sentence needs rewording. There's no clarity in what the authors want to convey, especially it been in the "Conclusions".

- pag. 12, line 401: When the authors say 'perform consistently'; do they mean 'consistently good' or 'consistently bad'?.

- pag. 13, line 412: Replace 'Last but not least, we would like to' by 'We also'.

- References section: there are at least two references in which the DOI links do not work, i.e., pag. 15, line 461; pag. 16, line 469. Please check that all the DOIs of the cited references point to valid addresses.

- pag. 18, line 515: Replace 'and "b"' by 'whereas "b"'. I also suggest a change in the color scale. It's hard to distinguish those CMLs in yellow.

– pag. 19, Figure 3: I still don't see the need for a 5-min moving average. This means that in principle, the authors are averaging out potential larger 1-min intensities, for instance. So, what's the purpose of using a 5-min moving average, if in the end the authors are performing analyses at 1-min resolution (e.g. Figure 11).

– pag. 20, Figure 4: I think these two sub-figures/panels could be join into only one figure, by having a secondary Y-axis (on the right) representing either rainfall intensity or attenuation

– pag. 22, line 555: Replace 'line $y = x$' by '1:1 line'.

– pag. 25, line 568: Replace 'Metadata table of CMLs' by 'CML metadata'.

– pag. 25, line 569: Remove the first sentence.

– pag. 25, lines 569 to 572: This paragraph should not be presented as an independent paragraph but as the caption of Table A1.

– pag. 25, line 570: Replace 'The frequency' by 'The frequencies'.

– pag. 25, line 571: Replace 'of one link' by 'in one link path'.

– pag. 26, line 574: Add this line add at the end of the caption of Table A1.

– pag. 25, Table A1: Remove the current title of the table. Column $\alpha - \beta$ should be split into two columns (i.e., $\alpha$, and $\beta$) (if that creates a wider table, then present it in landscape mode). Columns 'Antenna size node A (m)', and 'Antenna size node B (m)' are taking too much space, and not contributing too much to the table; you can avoid this just by stating (e.g., in the caption of the table) that all antenna sizes are $30\,\mathrm{cm}$, except for node B in site 1b (UKY 22046/SC15), and nodes B in sites 10a...etc.

– pags. 27 and 28, Table B1: I don't see the need of this table, especially given that the authors presented this data in Figure 3.

**REFERENCES**

[1] Song, K., Liu, X., Zou, M., Zhou, D., Wu, H., and Ji, F.: Experimental Study of Detecting Rainfall Using Microwave Links: Classification of Wet and Dry Periods, IEEE J. Sel. Topics Appl. Earth Observ. Remote Sens., 13, 5264–5271, https://doi.org/10.1109/JSTARS.2020.3021555, 2020.

---

## Author Comment (AC1)

**Responses to comments posted by Referee #1**

We thank the referee for reviewing our manuscript and providing constructive feedback. It helped us to clarify unambiguous phrasings and the presentation of the analysis. We answer all comments in the following text. Our answers are in blue.

6: it is not very clear whether the d and z coefficients have been estimated from a different set of events or from the same events used for the analysis.

True, we complemented the text.

The changes in the original text (lines 213 ff) are in red:

*… Reference rain rates were calculated as a mean of rain rates along a CML path weighted by a CML path length intersecting radar product grid cells. The parameters for all links are optimized to minimum root mean squared error (RMSE) between CML and radar rain rates at hourly timesteps. The optimization uses complete hourly observations from the same rainfall events as used in the analysis.*

Lines 276-277: please specify what you mean with link sensitivity and how it is estimated.

We suggest a more detailed explanation of the meaning of CML sensitivity in the second section *CML rainfall retrieval and its uncertainty* (line 130 ff in original manuscript), where sensitivity is mentioned for the first time.

The changes in the original text (line 130 ff) are in red:

*The magnitude of total rainfall estimation error and the ratio between its components is strongly related to CML sensitivity to raindrop path attenuation, which is given by CML path length, frequency, and polarization. Sensitivity is defined as a measure of change in raindrop path attenuation per change in rainfall intensity.*

Figure 6 – Caption: I guess the number of collocated pairs should be 8 and not 5.

Thank you for spotting this typo. We corrected this in a revised version of the manuscript.

Figures 7 and 9: the two figures look a little confusing. I suggest to color or tag them with respect to the different pairs. In this way they could also be better referred in the main text. Moreover, I think that it could be useful two split each figure into two, providing double-mass

curves of cumulative rain for stratiform and convective events separately. The starting and ending of each event could also be marked on the curves.

We introduced color-coding for individual sites. We also separated the event types in side-by plots while keeping the overall double-mass curves with distinguished CML pairs. However, indication of the starts of individual events in double mass curves did not work well. Having relatively good fit for all pairs along the diagonal and having different accumulated rainfall amounts at each site, the plots appeared to be too crowded by the event start indicators. This would work well for simple time-accumulated rainfall, but this is not the case.

The changes in the original text (line 282 ff) are in red:

*To explore the systematic component of the measurement deviations, double-mass curves of cumulative rain of the CML pairs are displayed (in Figure. 7 (left). The lines show the main direction of the curves parallel to the diagonal, which indicates synchronized systematic errors of the independent sensors. Distinguishing the convective and stratiform events does not highlight any change in overall good fit (Fig. 7 right). However, changes in the trend of systematic errors can be observed in the course of time. For example, by the pair with highest cumulative rainfall (site 10a), link 2 observes systematically lower rain rates than link 1 up to rainfall depths around 200 mm, however, this trend then changes resulting in very low relative error between overall cumulative rainfalls at the end of the observation period. Overall, the pairs have a relative error between 0.01 and 0.18.*

The new plots are below with changes in the original caption text in red:

[Figure]

*Figure 7: (left) Double-mass curve for collocated sensors operating at identical frequency bands. (right) The same, but the convective and stratiform rainfall types are separated.*

[Figure]

*Figure 9: (left) Double-mass curve for collocated sensors operating at different frequency bands.  (right) The same, but the convective and stratiform rainfall types are separated.*

Did you try to calibrate WAA parameter for every single link? In this way you would have the best WAA calibration for each link and maybe it could be possible to better identify the discrepancies in Fig. 9.

The approach of calibrating all CMLs at once comes from the cited publication of Pastorek et al. (2022), which demonstrated similar results for calibration separately for each CML and for all CMLs at once. Nevertheless, we have also tested individual CML calibration and the impact was negligible (there were both improvements and deteriorations mostly in order of 0.1 mm h$^{-1}$), thus the manuscript follows the data processing using calibration for all CMLs all at once.

Pastorek, J., Fencl, M., Rieckermann, J., and Bareš, V.: Precipitation estimates from commercial microwave links: Practical approaches to wet-Antenna correction, IEEE Transactions on Geoscience and Remote Sensing, vol.60, pp.1-9, https://doi.org/10.1109/TGRS.2021.3110004, 2022.

Authors could provide a table summarizing the uncertainties associated to the different sources of error (taken from literature) in order to compare them with the inherent error of rain-induced attenuation.

Investigation of the individual components of rain-induced attenuation and their uncertainties is out of scope of the manuscript. However, based on this comment we decided to shortly discuss it in the Discussion section. The rain-induced attenuation is estimated as the difference between total loss and baseline. The total loss consists of several components (Eq.

1 in the manuscript) and for their quantification and quantification of their individual uncertainties is required thorough understanding of atmospheric conditions along the path of each CML and knowledge of each sensor hardware sensitivity to such atmospheric conditions. The rain-induced attenuation error was found to be at similar magnitude as the signal quantization. Thus, the rain-induced attenuation error is minor compared to the other errors. We decided to put a brief discussion on this and its consequences in the section 5 *Discussion*.

The changes in the original text (line 345 ff) are in red:

**5 Discussion**

 *In general, the collocated CMLs are in excellent agreement. The rain-induced attenuation error of CML pairs operating at identical frequency bands is 0.4 dB, which is close to the signal quantization level. The random error in rain-induced attenuation is minor compared to the systematic errors. The excellent agreement between collocated CMLs of the same frequency shows that CML hardware under the same atmospheric conditions provides homogeneous measurements of rain induced attenuation.*

*The errors of collocated CMLs operating at different frequencies are larger. Relative errors in estimated rain rate depths are 0.12 - 0.24 compared to 0.01 - 0.18 by CMLs of identical frequency. The increase of the relative error can be partly explained by different sensitivity of k-R relation accuracy on variable DSD along a CML path. Berne and Uijlenhoet (2007) simulated this effect and reported that systematic errors in rainfall depths (compared to the true rainfall) are relatively insensitive to CML path length and, for frequency range used in this study, are decreasing with increasing frequency by approx. -0.7 mm h-1 per 1 GHz. For the collocated CMLs operating at different frequency bands, the frequency separation reaches 3 to 7 GHz, and one can thus expect DSD related bias in the range of 2 mm h-1 to 5 mm h-1. The rest of the relative error is probably attributed to the hardware and differences in WAA. As this "residual relative error" is in the similar range as by the collocated CMLs of the same frequency, we suppose that CML frequency has relatively low impact on the WAA magnitude and pattern. In general, all the collocated CMLs are affected by almost the same WAA.*

*The errors reported in this study are attributed to hardware inhomogeneity and have similar magnitude as the errors reported by the studies evaluating CML performance against independent reference in dedicated experiments having optimal WAA model. For example, a dedicated experiment with accurate reference along the CML path reported errors 1.4 mm h-1 to 0.7 mm h-1 (for constant WAA model to the optimal dynamic model) by 38 GHz CML mainly attributed to WAA (Schleiss et al., 2013). The errors reported in larger case studies are, however, higher. The measurement accuracy errors might be partly explained by larger hardware inhomogeneity (de Vos et al., 2019) However, substantial part of these errors is probably also attributed to the uncertainties in the reference measurements, which are difficult to quantify in large-scale evaluations and thus none of these studies explicitly accounts for them.*

*Other aspects affecting the performance of collocated CMLs, such as different hardware, size of the dataset and different complexity of the data processing, are discussed in the following subsections.*

**5.1 Effect of different hardware**

…

---

## Author Comment (AC2)

**Responses to comments posted by Referee #2**

We thank the second referee for reviewing our manuscript and providing insightful feedback. Several unclear issues were clarified. We also appreciate detailed minor comments, which helped to improve the manuscript. From minor comments we accepted the vast majority of the suggested stylistic changes. The manuscript was checked by native speaking proofreader and we followed the AMT manuscript preparation instructions. We answer all comments in the following text. Our answers are in blue.

**MAJOR COMMENTS**

– pag. 7, line 197: What the maximum 5-min intensity is computed for?. There seems to be no use of such estimates further down the analyses. Besides, further ahead in line 198, the median of the maximum 5-min intensities is not shown in Figure 3 (as the authors seem to suggest).

Thank you for highlighting this ambiguousness. We will be more precise in our statement. The data for Fig. 3 come from the 23 municipal rain gauges, as stated. We decided to use the rain gauge data only for the plot of event event characteristics as the rain gauges provide direct measurement on the ground at 1-min temporal resolution.

The median is mentioned only in the text, therefore, we described Figure 3 in a separate sentence.

The changes in the original text (lines 197 ff) are in red:

*From a visual inspection of weather radar images and analysis of rain gauge rainfall intensities, events were divided into two groups: One, convective rainfall events characterized by higher intensities, short durations, and a spatially limited area. The other, stratiform rainfall events with lower intensities, longer duration, and a more extensive area. The maxim 5-min rainfall intensity was calculated for each of the 23 municipal rain gauges, as well as the median of these intensities . Figure 3 shows the maximum 5-min rainfall intensity of the 23 municipal rain gauges for each event and the event durations. The median maxim 5-min intensity during convective events was 68 mm h$^{-1}$ and during stratiform events was 21 mm h$^{-1}$. The median durations of convective and stratiform events was 4 h (in the range of 1 to 18 h), resp. 12 h (in the range of 5 to 34 h). There were 10 stratiform events and 23 convective events, of which 21 were in the spring and summer seasons.*

– pag. 8, lines 219 to 227: Enclose these two paragraphs into a section titled 'Radar data', for instance.

Having in mind the minor role of the radar data in the storyline, we decided to reduce the paragraphs dedicated to the radar data processing in section *3.3 Processing of raw CML data* (removing lines 219 to 224). We would prefer to keep the information about the radar

adjustment in the material (replacement of lines 225 to 227 as a new last paragraph in *3.1 Materials*, subsection *Gauge-adjusted radar rainfall*).

– pag. 8, line 231; and pag. 9, line 246: Equations (8) and (9) are very basic equations that can be removed, given that they are not essential in the analyses nor in the results presented in this manuscript.

Agreed.

– pag. 8, lines 232 to 241; and pag. 11-12, lines 333 to 341, pag. 24, Table 1: The authors propose 3 alternative methods to estimate baseline and WAA. Nevertheless, the description of such methods is not entirely clear. Furthermore, the authors do not offer a solid argument (neither do they present compelling results/figures) on how these alternatives would benefit potential future studies (which apparently is not the case, given what they conclude in pag. 12, lines 342 to 344). Overall, this unclarity and lack of results leads to confusion, and aimlessness with regard to alternatives in baseline and WWA estimates. What I suggest is that the authors either offer a more in-depth view of the alternatives they present, and the benefits in implementing them (with results included); or do not mention/present at all such alternatives, thus avoiding obscuring the simple/basic idea they center their work around. If the authors proceed with presenting these 3 alternatives, the enumeration (of the alternatives) should be 1 to 4, instead of 0 to 3 (as it presently is). It also worth mentioning in your description/analyses of Wet/Dry classification periods, works such as Song et al. [1], for instance.

Thank you for this comment, you are right. The description of the alternative processing methods is not comprehensive as our focus was on the analysis of collocated CMLs more than the alternative processing methods. Therefore, we deal with our standard processing protocols in this study and we only test if the results are sensitive to complexity of processing chain. As we observed that the increasing complexity can affect the performance of collocated CMLs, we believe that information about possible consequences of the choice of the processing should be mentioned.

To conclude, we suggest keeping focus of the study on the core of the work, which is the performance of the collocated sensors using the processing which performs the best to our knowledge and replacing the different processing methods and their performance to a new subsection in Discussion (*5.3 Effect of different processing methods*). We would also move the lines 232 to 241 of the original manuscript describing the other processing methods to the new subsection in the Discussion.

– pag. 11, line 323: Instead of analyses for 10 − min aggregations, I'd prefer to see aggregations at 1 − h. In practical applications 5 and 15 − min aggregations are near enough

(to 10 − min). Nevertheless, having hourly aggregates/estimates would allow comparisons with other rainfall products (e.g., satellite).

Agreed. We have changed the figure and modified the text in the manuscript accordingly. The changes in the original text (lines 197 ff) are in red:

*RMSE and correlation were calculated for time aggregations of 1, 5,  15,  30 and 60 min (Fig. 11) for both rainfall types (convective and stratiform) and for both types of sites (sites "a" including the sites with identical frequency band CMLs, and sites "b" with CMLs with different frequency bands). The aggregation brings a reduction of noisiness. The correlation coefficient is consistently greater than 0.92. The increase in correlation is more pronounced for the time aggregation between 1 and 5 min. Further aggregation did not bring any additional improvement in this performance measure. The agreement of devices expressed as RMSE improves particularly quickly for convective rainfall types in aggregation between 1 and 1 min from RMSE 0.9 mm h⁻¹ to 0.6 mm h⁻¹ for pairs of identical frequency bands, and from 1.3 mm h⁻¹ to 1.1 mm h⁻¹ for pairs of different frequency bands. Stratiform rainfall types do not rapidly improve in RMSE with greater time aggregation, but the enhancement occurs linearly with greater time aggregation. On the other hand, for convective rainfall types improvement accelerates, and  it is even more pronounced for CML pairs at an identical frequency band.*

The new plot is below with changes in the original caption text in red:

[Figure]

*Figure 11: Influence of time aggregation (1, 5,  15,  30 and 60 min) on (left) correlation and (right) RMSE in mm h⁻¹.*

– pag. 19, Figure 2: This would be a more valuable plot if it were presented as a scatterplot of Frequency vs. Length (in the style of Figure 2, of your reference in line 420; or Figure 7, of your reference in line 428; or Figure 2, of your reference in line 443); which it is more common and "standard" in CML studies. If any color scale is to be applied, that could be site-code, for instance.

Originally, the CMLs at sites were plotted as you suggest. However, having many CMLs with similar lengths (e.g. sites 3a,b, 5b, 6a, 8a, 10a all around 1000 m long, and similarly, 1a, 2a,b,

9a, 11b all around 2500 m) we ended up with many points on top of each other and even color scale for site-coding and subplots zooming in the area did not help. Such conditions made distinguishing the individual sites impossible.

The goal of Figure 2 is not to provide an overall metadata of the CML frequencies and lengths and thus link the plot to their sensitivity. In our study on collocated sensors, we prefer to use a format where the information related to one site is as concentrated/collocated as possible and where the site-related differences are highlighted.

– pag. 20, Figure 5; and pag. 21, Figure 6B; and pag. 22, Figure 8B: Join these figures into one (i.e., one figure three panels).

Thank you for the suggested layout of the figures. However, we would like to keep the storyline of the Results section dedicated, first, to the CML pairs operating at identical frequency bands, and second, to the CML pairs operating at different frequency bands. Joining Figures 6B and 8B would cause discontinuous figure references in the text and thus would be in disagreement with AMT manuscript preparation guidelines, which requires the consecutive numbering of figures. Another option would be to completely restructure the paragraphs, but this is not our preference as stated above.

Joining Figures 5 and 6 would mix the plots of rain-induced attenuation and derived rainfall intensity, which we also find as an inconvenient solution.

We think that this comment (and the next two), are mostly related to the author's writing, stylistic and presentation style, and also the storyline of the paper. Of course, the proposed changes could also be an option, but in this case, we would like to keep it as presented.

– pag. 21, Figure 6A; and pag. 22, Figure 8A: Join these figures into one (i.e., one figure two panels).

Similarly as the previous, the consecutive numbering of figures would not be compiled.

– pag. 21, Figure 7; and pag. 22, Figure 9: Join these figures into one (i.e., one figure two panels).

Similarly as one before the previous, the consecutive numbering of figures would not be compiled.

**MINOR COMMENTS**

– pag. 1, line 10: Replace 'reference' by 'reference to'.

Changed.

– pag. 1, line 13: Replace '2014 and 2016' by '2014 - 2016' (given that your study goes from 2014 through 2016, right?).

Changed.

– pag. 1, line 13: Replace 'in 1-min' by 'at 1-min'.

Changed.

– pag. 1, lines 14, and 15: Replace 'commercial microwave links' by 'CMLs' (from line 1 the authors established the acronym, so why not use it here too?).

Replaced.

– pag. 1, line 27: Replace 'studies' by 'studies have'.

Replaced.

– pag. 2, line 27: Replace 'have not been, to the best of our knowledge, reported' by 'have not been reported, to the best of our knowledge'.

Your comment is probably linked to line 59, where we changed the sentence.

– pag. 3, line 64: Replace 'or' by 'and'.

Replaced.

– pag. 3, line 68: Replace 'the same frequencies and at' by 'equal and'.

We prefer to keep "the same frequencies" as it is a less strict expression than "equal", which suggests the mathematical operator "=", which is not true in our case of collocated CMLs operating at the same frequency band (i.e., not identical/equal frequencies).

– pag. 3, line 69: Replace 'these' by 'such'.

Replaced.

– pag. 3, line 75: Remove 'finally,'.

Removed.

– pag. 3, line 77: Replace 'utilizes' by 'uses'.

Replaced.

– pag. 3, line 78: Replace 'total loss' by 'total path loss'.

Replaced.

– pag. 3, line 79: Replace 'rainy' by 'wet'.

Replaced.

– pag. 3, line 79: Replace 'and accounting for' by 'plus the'.

We suggest keeping the word "accounting" rather than "plus" because "plus" would suggest that the wet antenna attenuation is summed up with the observed total path loss in the processing.

– pag. 3, line 79: Replace 'that is caused by water' by 'caused by the water'.

Replaced.

– pag. 3, line 80: Replace 'forming on antenna' by 'accrued on the antenna'.

We suggest keeping the word "forming" rather than "accrued" because "accrued" would suggest that the raindrops are accumulating on the antenna covers and are permanently present on the antenna covers.

– pag. 3, line 80: Replace 'rain and dew' by 'rain and/or dew'.

Replaced.

– pag. 3, line 80: Add your reference 'Leth et al., 2018' (which should actually be 'van Leth et al., 2018') after/before 'Chwala and Kunstmann, 2019' (please, note that is 'Chwala', and not 'Chawla'). Double check all your references for such inconsistencies.

Thank you for spotting this typo, the reference was added.

– pag. 3, line 80: 'The specific raindrop attenuation...'. You have so far not defined what 'specific attenuation' is. Please, add accordingly something like: ', which is the total path loss divided by the link-path distance...'.

True, we added the relative clause explaining the definition of specific attenuation.

– pag. 3, line 81: Replace 'estimation' by 'estimates'.

Replaced.

– pag. 3, line 91: Replace '0.3 resp. 1 dB' by '0.3 and 1 dB, respectively'. Furthermore, the authors seem to use (quite often) 'resp.' as an acronym of 'respectively', something I have personally never seen before. Therefore, please change accordingly all the 'resp.' instances (e.g., lines 97, 186, 200).

Replaced, however, the "resp." abbreviation seems to be used quite frequently in other texts.

– pag. 3, line 92: Please specify again what the 'second case' refers to.

Thank you, we will be more specific.

– pag. 4, lines 98, and 100: Remove 'losses due to'.

Removed.

– pag. 4, line 99: Replace 'losses causing' by 'caused by the'.

Replaced.

– pag. 4, line 100: Remove 'wet antenna attenuation' (The authors already established the acronym in pag. 3).

Removed.

– pag. 4, lines 101, and 102: Remove this sentence. The authors already established what WAA is (– pag. 3, line 80).

We prefer to keep the sentence as it is a more descriptive explanation of the process with mentioning the factors, which influence it.

– pag. 4, line 103: Replace 'separated' by 'regard to'.

We would like to use the word "separated" as it clearly explains the mathematical operation that needs to be used.

– pag. 4, line 106: Remove ', thus,'.

Removed.

– pag. 4, line 108: Replace 'necessarily fulfilled' by 'entirely accurate'.

Replaced.

– pag. 4, line 109: Replace ', as well as temperature, can' by 'and temperature'.

Replaced.

– pag. 4, line 110: I'd suggest replacing 'raindrop path attenuation' by 'attenuation from raindrops'; in all possible instances, e.g., lines 131, and 134.

We prefer to keep "raindrop path attenuation" as it empathizes the path-integrated character of the observation.

– pag. 4, line 111: Replace 'can' by 'do'.

We just removed the word "can".

– pag. 4, line 119: Replace 'WAA magnitude and pattern' by 'patterns of WAA magnitude'.

Replaced.

– pag. 4, line 121: Replace 'a power' by 'the power'.

Replaced.

– pag. 5, line 124: Replace 'on CML' by 'on the CML'.

Replaced.

– pag. 5, line 125: Use the DSD acronym first in pag. 2, line 36 (not here).

Agreed.

– pag. 5, line 125: Replace 'alpha and beta' by 'α and β'.

Replaced.

– pag. 5, line 128: Replace 'are lowest' by 'are the lowest'.

Replaced.

– pag. 5, line 130: Replace 'of total' by 'of the total'.

Replaced.

– pag. 5, line 131: Replace 'by CML' by 'by the CML'.

Replaced.

– pag. 5, line 132: Replace 'incorrectly estimated' by 'wrong estimates of'.

We would keep the "incorrectly estimated" as "wrong estimates of" would suggest that the application of the WAA method was "wrongly applied with wrong estimates" not that the applied method was fine with inaccurate estimates.

– pag. 5, line 133: Replace 'k − R relation non-linearity' by 'the non-linearity of the k − R relation'.

Replaced.

– pag. 5, line 137: Replace 'The same collective of authors' by 'Fencl et al., 2020 also' (and remove this reference at the end of the sentence).

Replaced.

– pag. 5, line 139: Remove 'alone'.

We suggest reformulating to "especially in cases where CML observations are a single available rainfall observation data source" instead.

– pag. 5, line 142: Remove 'can, however,'.

We prefer to keep the word there as it highlights the contrast.

– pag. 5, line 143: Remove 'Losses on the antennas also include WAA caused by the formation of a water layer on the antenna radomes.' (too repetitive by now).

Agreed.

– pag. 5, line 147: Replace 'EM' by 'Eletromagnetic (EM)' (this acronym was not previously established).

"EM" is established at line 77.

– pag. 5, line 148: Replace ', however, they interact' by 'interacting'.

Replaced.

– pag. 5, line 148: Replace 'Discrepancies' by 'Therefore, discrepancies'.

Replaced.

– pag. 5, line 149: Remove 'thus'.

Removed.

– pag. 5, line 150: Replace 'the same frequency' by 'equal'.

We prefer to keep "the same frequency" as it is a less strict expression than "equal", which suggests the mathematical operator "=", which is not true in our case of collocated CMLs operating at the same frequency band (i.e. not identical/equal frequencies).

– pag. 5, lines 152 to 154: Remove the last sentence (it's repetitive of the above sentence, and adds confusion instead of clarity).

Removed.

– pag. 6, line 156: Replace 'twelve' by '12'.

Replaced.

– pag. 6, line 160: Remove 'one pair of' (in both instances).

Removed.

– pag. 6, line 161: Replace 'intercompared for' by 'compared among'.

Replaced.

– pag. 6, line 151: Replace 'frequency bands' by 'frequencies' (here and everywhere else in the manuscript; e.g., lines 169, 244, 247, 263, 264, 288, 289, 382, 385, 386, 515, 570).

We prefer to keep "frequency bands" as we do not refer to the CMLs operating at a certain frequency, but we categorized them into standardized frequency bands.

– pag. 6, line 162: Replace 'sites' by 'the sites'.

Replaced.

– pag. 6, line 164: Remove the title 'Materials'.

We do not see the benefit of removing the title as it really covers the description of the material used in the study.

– pag. 6, line 165: Enumerate 'CML' as 3.1.

As previous.

– pag. 6, line 168: Replace 'To compare, such' by 'Such'.

Replaced.

– pag. 6, line 169: Replace 'urbanized' by 'urban'.

Replaced.

– pag. 6, line 170: Replace '682 m to 5836 m' by '~ 0.7 to ~ 5.8 km'.

Replaced.

– pag. 6, line 173: Replace 'having' by 'having a'.

Replaced.

– pag. 6, line 173: Replace 'are 0.3 m or 0.6 m in diameter' by 'have diameter of 30 or 60 cm'.

Replaced.

– pag. 6, line 174: Replace 'polarizations, are' by 'polarization is'.

Replaced.

– pag. 6, line 176: Replace 'respectively' by ', respectively'.

Replaced.

– pag. 6, line 177: Enumerate 'Gauge-adjusted radar rainfall' as 3.2 (and re-enumerate the remaining titles accordingly).

Same as 3.1

– pag. 6, line 180: Replace 'permanent municipal RG' by 'municipal rain gauge' (as the authors do not use anywhere else the 'RG' acronym).

Replaced.

– pag. 6, line 182: Replace 'monthly and regularly dynamically calibrated' by 'and regularly calibrated on a monthly basis'.

It would change the meaning of the sentence. The rain gauges are not calibrated on a monthly basis.

– pag. 6, line 186: Replace 'spatial and temporal' by 'spatiotemporal'.

Replaced.

– pag. 6, line 186: Replace 'resp.' by 'every'.

Replaced.

– pag. 7, line 188: Replace 'uses a data set collected during the monitoring of the' by 'is for the monitoring'.

We believe that it would change the meaning of the sentence. We present only a small part of the collected data.

– pag. 7, line 188: Replace '2014 and' by '2014 through'.

Replaced.

– pag. 7, line 189: Replace 'In total' by 'In total,'.

Replaced.

– pag. 7, line 190: Replace 'one hour' by '1 h'.

Replaced.

– pag. 7, line 191: Replace '304 h and 15 min' by '~ 304 h'.

Replaced.

– pag. 7, line 191: Remove 'and 15 min'.

Removed.

– pag. 7, line 194: Replace 'First' by 'one'.

Replaced.

– pag. 7, line 195: Replace 'Second' by 'the other'.

Replaced.

– pag. 7, line 196: Replace 'ten' by '10'.

Replaced.

– pag. 7, line 197: Replace 'maximal' by 'maximum'.

Replaced.

– pag. 7, line 198: Replace 'median maximal' by 'average maximum'.

It is really median not average. We used median as it better represents the event characteristics.

– pag. 7, line 199: Replace 'reached up to' by 'was'.

Replaced.

– pag. 7, line 199: Replace 'events up to' by 'events was'.

Replaced.

– pag. 7, line 199: Replace 'median' by 'average'.

It is really median not average. We used median as it better represents the event characteristics.

– pag. 7, line 202: Remove 'temporal'.

Removed.

– pag. 7, line 203: It is not clear how the authors perform 1-min aggregations 'using averaging'.

We will be more precise in this sentence. The data acquisition software is described in the Material, where it is mentioned that it polls the data approximately every 10 s. We added "of the polled data (~10 s)" at the end of the sentence.

– pag. 7, line 210: Replace 'wet antenna attenuation model was' by 'WAA model is'.

Replaced.

– pag. 7, line 213: Replace 'maximum Aw' by 'maximum WAA' (why not use WAA instead of Aw in Equation (6)?).

We prefer "Aw" as in all equations the abbreviations operating with attenuation (dB) start with the capital "A".

– pag. 8, line 216: Replace 'product grid cells' by 'pixels'.

Replaced.

– pag. 8, line 225: Replace 'The data were resampled to 1 h timesteps' by 'Radar data was aggregated from 5-min to 1-h'.

Replaced.

– pag. 8, line 230: Remove 'of the' from 'mean of the rainfall'.

Removed.

– pag. 8, line 243: Remove '(Figure 2)'.

Removed.

– pag. 9, line 250: Replace ':' by ', i.e., '.

Replaced.

– pag. 9, line 252: Replace 'which one' by 'which at least one'.

Replaced.

– pag. 9, line 253: Remove the first sentence.

We would prefer to keep the sentence there as it explains what kind of data relation of the observation the metric reflects.

– pag. 9, line 256: Please clarify what do you mean by 'the deviation' (deviation from what?).

True, we replaced "the deviation" by "the difference", which better reflects what the metrics measure.

– pag. 9, line 258: Remove the last sentence (of the paragraph).

We would prefer to keep the sentence there as it explains what kind of data relation of the observation the curves reflect.

– pag. 9, lines 261 to 262; and pag.12, line 346: Please, remove this "filling" text (no need for it).

Removed.

– pag. 9, line 264: Remove 'are evaluated in this subsection'.

Removed.

– pag. 9, line 265: Replace 'demonstrates' by 'shows'.

We would keep "demonstrates" as the verb "show" would be used again in the following sentence.

– pag. 9, line 265: Remove 'at two stages of processing: rain-induced attenuations are shown in the top panel, and rain rates in the bottom panel'.

Removed.

– pag. 9, line 268: Remove ', which, on the other hand, is not evident from the scatterplots (Fig. 6)'.

Removed.

– pag. 9, line 269: Replace 'represents all eight' by 'shows all 8'.

We suggest using "shows all eight CML pairs" as it follows the AMT manuscript preparation guidelines ("For items other than units of time or measure, use words for cardinal numbers less than 10").

– pag. 9, line 272: Replace 'To conclude, this' by 'This'.

Replaced.

– pag. 9, line 273: Replace 'is low in total CML observation error' by 'in total CML observation error is low'.

Replaced.

– pag. 9, line 274: Replace 'Scatterplots' by 'Figure 6 shows scatterplots'.

Replaced. Moreover, we removed the figure reference at the end of the sentence.

– pag. 10, line 276: What is 'sensitivity'?; and how it's measured? (do you mean 'uncertainty', maybe?).

Thank you for the comment, similar question raised also in the first referee manuscript evaluation. Based on that we decided to add a more detailed description of the term *sensitivity* in section CML rainfall retrieval and its uncertainty.

– pag. 10, line 283: Replace '(Fig. 7)' by 'in Figure 7'.

Replaced.

– pag. 11, line 310: Remove ', thus,'.

Removed.

– pag. 11, line 315: When the authors say 'the WAA model used', to what WAA-model exactly you're referring to? (please see my "major" comment in regard to the models of WAA (and baseline) estimation).

Reflecting the major comment regarding different processing methods, this minor comment is solved.

– pag. 11, line 311: What do you mean by 'match'.

We suppose that you might refer to line 320. We replaced the word "match" by "agreement", which better represents the core of the message.

– pag. 11, line 332: Remove 'further,'.

Removed.

– pag. 12, lines 358, and 359: It is not clear form this last sentence whether the antenna size does influence (or not) the performance the rainfall retrievals. Please clarify.

As the site 12b has almost twice as long CML path as the other sites, we could not conclude whether the difference is caused by its path length (sensitivity) or size of the antenna radome.

We added a sentence "From the material used, it cannot be concluded whether the difference is caused by different CML path length (sensitivity) or the different antenna radome." at the end of the paragraph.

– pag. 13, line 376: Replace 'The presented' by 'This'.

Replaced.

– pag. 13, line 378: Replace 'The' by 'This'.

Replaced.

– pag. 13, line 385: Replace 'RMSE (differences) of the retrieved rainfall information' by 'differences in RMSE of rainfall retrievals'.

We prefer to keep the wording as the values of the metric do not represent the differences of RMSE among the sites, but the maximum and minimum RMSE among the sites. The

"(differences)" is there to demonstrate that we do not evaluate against reference (errors) but two sensors against each other (differences).

– pag. 13, line 386: Please update all instances of mm h$^{-1}$ to mm · h$^{-1}$.

The AMT manuscript preparation instructions do not use interpunct.

– pag. 13, line 388: Replace 'types,' by 'types (convective and stratiform),'.

Replaced.

– pag. 13, line 393: Remove this first sentence (please also see my "major" comment in regard to the models of WAA (and baseline) estimation).

Reflecting the major comment, this minor comment is solved.

– pag. 11, line 397: Remove ', first and foremost,'. This sentence needs rewording. There's no clarity in what the authors want to convey, especially it been in the "Conclusions".

The changes in the original text (lines 397 ff) are in red:

*Even though CMLs  have not been primarily deployed for rainfall monitoring, this study proved that such collocated opportunistic rainfall  sensors are in excellent agreement and the hardware is homogeneous in its behaviour.*

– pag. 12, line 401: When the authors say 'perform consistently'; do they mean 'consistently good' or 'consistently bad'?.

Good point. Reflecting the results, we wrote "consistently good".

– pag. 13, line 412: Replace 'Last but not least, we would like to' by 'We also'.

Replaced.

– References section: there are at least two references in which the DOI links do not work, i.e., pag. 15, line 461; pag. 16, line 469. Please check that all the DOIs of the cited references point to valid addresses.

The reference at line 461 accidentally contains the pages, which we removed in the revised manuscript. The reference at line 469 works fine for us and we did not find where the problem could be.

– pag. 18, line 515: Replace 'and "b"' by 'whereas "b"'. I also suggest a change in the color scale. It's hard to distinguish those CMLs in yellow.

Replaced. We changed the hue for the highest frequency band and also in Figure 2 to keep the color scale the same. Please see the figures below.

[Figure]

– pag. 19, Figure 3: I still don't see the need for a 5-min moving average. This means that in principle, the authors are averaging out potential larger 1-min intensities, for instance. So,

what's the purpose of using a 5-min moving average, if in the end the authors are performing analyses at 1-min resolution (e.g. Figure 11).

We will be more precise in our statement in the main text. Please see our response in the first major comment.

– pag. 20, Figure 4: I think these two sub-figures/panels could be join into only one figure, by having a secondary Y-axis (on the right) representing either rainfall intensity or attenuation

Thank you for the suggestion. Please see the figure and its caption below.

[Figure]

*Figure 4: Example of a time series of collocated independent sensors  at site 2a for () rain-induced attenuation and () rainfall intensity on 21$^{st}$ July 2014.*

– pag. 22, line 555: Replace 'line y = x' by '1:1 line'.

Reflecting the comment of the first reviewer, the sentence will not be in the revised manuscript.

– pag. 25, line 568: Replace 'Metadata table of CMLs' by 'CML metadata'.

Replaced.

– pag. 25, line 569: Remove the first sentence.

Removed.

– pag. 25, lines 569 to 572: This paragraph should not be presented as an independent paragraph but as the caption of Table A1.

Replaced.

– pag. 25, line 570: Replace 'The frequency' by 'The frequencies'.

Replaced.

– pag. 25, line 571: Replace 'of one link' by 'in one link path'.

Replaced.

– pag. 26, line 574: Add this line add at the end of the caption of Table A1.

Replaced.

– pag. 25, Table A1: Remove the current title of the table. Column α − β should be split into two columns (i.e., α, and β) (if that creates a wider table, then present it in landscape mode). Columns 'Antenna size node A (m)', and 'Antenna size node B (m)' are taking too much space, and not contributing too much to the table; you can avoid this just by stating (e.g., in the caption of the table) that all antenna sizes are 30 cm, except for node B in site 1b (UKY 22046/SC15), and nodes B in sites 10a...etc.

True, the table is adapted in the revised manuscript. We suggest removing the antenna types as we do not discuss them in the text.

– pags. 27 and 28, Table B1: I don't see the need of this table, especially given that the authors presented this data in Figure 3.

Table is removed in the revised manuscript.

**REFERENCES**

[1] Song, K., Liu, X., Zou, M., Zhou, D., Wu, H., and Ji, F.: Experimental Study of Detecting Rainfall Using Microwave Links: Classification of Wet and Dry Periods, IEEE J. Sel. Topics Appl. Earth Observ. Remote Sens., 13, 5264–5271, https://doi.org/10.1109/JSTARS.2020.3021555, 2020.

---

## Author Response (AR2)

**Responses to suggestions for revision posted by Referee #2**

We thank the second referee for reviewing our manuscript and for the suggested technical corrections. We answer the comments in the following text. Our answers are in blue.

**SUGGESTIONS FOR REVISION**

-- pag. 5, line 147 (of the 1st submitted revision): The accronym 'EM' was indeed previously established. It did escape my review. My apologies for that.

No problem.

-- Fig. 4 (updated): Please remove the number "-10" that appears on the right y-axis of the figure. If not removed, it may imply that negative precipitation can happen, which actually is not the case (even though the blue lines never drop below zero).

Thank you for highlighting this detail.

The figure in the revised manuscript was changed with respect to the comment, please see the updated figure below:

[Figure]